



# Regionally optimized fire parameterizations
# using feed-forward neural networks
Yoo-Geun Ham[1], Seung-Ho Nam[1], and Jin-Soo Kim[2]
[1]Department of Oceanography, Chonnam National University, Gwangju, 61186, South Korea
[2]Low-Carbon and Climate Impact Research Centre, School of Energy and Environment, City
University of Hong Kong, Tat Chee Ave, Kowloon Tong, Hong Kong, People's Republic of China
*Correspondence to*: Prof. Yoo-Geun Ham (ygham@chonnam.ac.kr)
The fire weather index (FWI) is a widely used metric for fire danger based on
meteorological observations. However, due to its empirical formulation based on a
specific regional relationship between the meteorological observations and fire
intensity, the ability of the FWI to accurately represent global satellite-derived fire
intensity observations is limited. In this study, we propose a fire parameterization
method using feed-forward neural networks (FFNNs) for individual grids. These
FFNNs for each grid point utilize four daily meteorological variables (2-meter relative
humidity (RH2m), precipitation, 2-meter temperature, and wind speed) as inputs. The
outputs of the FFNNs are satellite-derived fire radiative power (FRP) values. Applying
the proposed FFNNs for fire parameterization during the 2001–2020 period revealed a
marked enhancement in cross-validated skill compared to parameterization solely
based on the FWI. This improvement was particularly notable across East Asia, Russia,
the eastern US, southern South America, and central Africa. The sensitivity
experiments demonstrated that the RH2m is the most critical variable in estimating the
FRP and its regional differences via the FFNNs. Conversely, the FWI-based
estimations were primarily influenced by precipitation. The FFNNs accurately captured
the observed nonlinear correlations between FRP and RH2m, as well as precipitation.
In contrast, FWI-based estimations exhibit an excessively negative relationship
between FRP and precipitation.

**Keywords**: fire parameterization, fire radiative power, fire weather index, feed-
forward neural networks



## 1. Introduction

Fires are inflicting substantial ecological and socio-economic impacts on a global scale. Furthermore, monitoring and managing the risk of fire incidents at an early stage poses a significant challenge for each country (Vitolo et al., 2019). Fire ignitions encompass both natural factors such as lightning, as well as human activities (Jones et al., 2022). After ignition, fire propagation is mainly determined by dryness (Bistinas et al., 2014, Abatzoglou and Williams 2016). Spatially estimating and forecasting dryness enables the monitoring of fire hazards, thus facilitating the implementation of emergency measures to curb the expansion of uncontrollable large fires (Di Giuseppe et al., 2016, Bett et al., 2020, Haas et al., 2022).

Among several operational fire danger indices, the fire weather index (FWI) holds a prominent status as an indicator of potential fire intensity. Developed by the Canadian Forest Fire Danger Rating System (Van Wagner 1974, 1987), the FWI is based on four daily meteorological observations: near-surface air temperature, near-surface air relative humidity, wind speed, and precipitation. Fuel moisture codes are first determined from meteorological data to assign numerical ratings to the moisture content of the forest floor and other deceased organic matter. Afterward, fire behavior indices (initial spread index and buildup index) are calculated based on a combination of meteorological observations and fuel moisture codes. Finally, these indices are used to calculate the FWI, thus providing an estimation of fire intensity (Vitolo et al., 2019).

Although this system has been shown to be globally applicable (Bedia et al., 2015, Abatzoglou et al., 2018), it was originally developed for the characterization of evergreen pine stands in forested areas of Canada. Therefore, all links between fire moisture codes and fire behavior indices are optimized and parameterized for eastern Canada. However, regional fire dynamics vary significantly depending on vegetation species and their distribution, climatological seasonal cycle, and biogeomorphological characteristics (Ducan et al., 2003, Flannigan et al., 2005, Macias Fauria et al., 2011, Rogers et al. 2015, Kim et al., 2019). For instance, extensive deforestation fires in the Amazon are attributed to insufficient cumulative precipitation (Le Page et al., 2010), whereas Arctic fire activity is more sensitive to temperature (Kim et al., 2020). As reported by Grillakis et al., (2022), each meteorological observation in the FWI system holds varying sensitivity to remotely sensed fire activity.

To calibrate the varying sensitivities of the FWI system to meteorological parameters and obtain accurate estimations of fire activity, our study optimized fire





parameterizations with satellite-derived fire radiative power (FRP) datasets based on
feed-forward neural networks (FFNNs) in each region with fire activity records. Given
that FFNNs follow the same structure and input variables as the FWI, the parameter
values linking meteorological observations, fire moisture code, and fire behavior
indices are established for every 1° × 1° resolution grid box via FFNNs, thus foregoing
raw parameterizations in the Canadian FWI. In addition to our novel FFNN-based
model, we also conducted an in-depth examination of the FWI-based linear regression
model with FRP for comparative purposes. To quantify the relative contributions of
each meteorological parameter to the fire parameterizations, sensitivity experiments
were conducted based on climatological values of meteorological observations.

**2. Data and Experimental Design**
2.1. Data
2.1.1. Fire radiative power (FRP)
Given that the FWI was designed to estimate potential fire intensity, our analyses were
based on satellite-derived FRP, a metric that represents the rate at which a fire emits
energy in the form of thermal radiation. Specifically, daily FRP data was sourced from
the Moderate Resolution Imaging Spectroradiometer (MODIS) Collection 6.1 dataset
provided by the Fire Information for Resource Management System (FIRMS)
(https://firms.modaps.eosdis.nasa.gov/active_fire/) (Giglio et al., 2016). The period of
the FRP data spans from 2001 to 2020. The dataset featured a spatial resolution of 1°×1°
across the entire globe (0°–360°E, 90°S–90°N), with values expressed in megawatts
($10^6$ J s$^{-1}$; MW). It is important to note that although products were generated for both
land and ocean areas, we exclusively focused on land values, as FRP is directly
associated with fire size and intensity over terrestrial surfaces.

2.1.2. Meteorological observations
Meteorological observations are required as an input of the FWI and the FFNNs for the
FRP parameterizations. In this study, we used daily 2 m air temperature (T2m), 2 m air
relative humidity (RH2m), 10 m wind speed (WS10m), and precipitation (PRCP) from
ERA5 reanalysis produced by the European Centre for Medium-Range Weather
Forecasts (ECMWF) from 2001 to 2020 (Hersbach et al., 2020). The original horizontal
resolution was a quarter degree but was interpolated to a 1°×1° resolution over the





entire globe (0°–360°E, 90°S–90°N).
## 2.2. Models
### 2.2.1. FWI-based linear regression model
A linear regression model was established as a baseline for estimating fire occurrences
based on the meterological variables. This model takes the FWI as its input and yields
the FRP as its output. The linear regression coefficient was separately determined for
each grid point. The source code to produce the FWI was obtained from the Canadian
Forest Service at https://cfs.nrcan.gc.ca/publications/download-pdf/36461. A cross-
validation strategy was adopted for the skill assessment. For more details, please refer
to section 2.3.

### 2.2.2. FFNNs for FRP parameterization
The FFNNs employed for FRP parameterization consist of one input layer, three hidden
layers, and one output layer (Supplementary Fig. S1). The input layer comprises four
neurons corresponding to daily measurements of T2m, RH2m, WS10m, and PRCP at
a specific grid point. The output layer, on the other hand, encompasses a single neuron
responsible for concurrent FRP estimation at the corresponding grid point. Notably,
FFNNs are configured individually for each grid point. The first, second, and third
hidden layers are composed of 64, 32, and 16 neurons, respectively. Activation
functions are implemented utilizing the ReLU function. Techniques such as batch
normalization and dropout, with a dropout rate of 0.2, are applied to enhance model
robustness. It should be noted that the meteorological observations serving as input for
the FFNNs mirror those employed in the FWI. Thus, any disparities in estimation
accuracy between the FFNNs and the FWI-based model solely stem from the FRP
estimation algorithm.
The loss function of the FFNNs is defined as the root-mean-squared difference
between the observed FRP (y) and the estimated FRP (ŷ) as follows.
$$\text{Loss} = \sum_{i=1}^{N} (y_i - \hat{y}_i)^2$$

where N denotes the number of training samples. Similar to the FWI-based model, a
cross-validation strategy is adapted for the skill assessment (see section 2.3 for more
details).





2.3. Experimental design
The performance of both the FFNNs and the FWI-based linear regression model was
assessed by adopting a cross-validation strategy. The dataset was partitioned into
distinct subsets for testing, validation, and training purposes. The testing period was
defined by dividing the entire period from 2001 to 2020 into three-year intervals. The
validation dataset is defined as the last two years of each three-year interval, whereas
the remaining data was used for training. For example, for the 2001–2004 test period,
the models were trained using a 2005–2018 dataset, whereas the data from 2019–2020
was used for validation. Additional details on the selection of periods for training,
validation, and testing are provided in Supplementary Table S1. Next, the FRP was
estimated using both FFNNs and FWI-based linear regression models across the 2001–
2020 period. To assess the FRP estimation accuracy of the evaluation procedures, FRP
anomalies were calculated by subtracting the estimated daily climatology throughout
the 2001–2020 period.

**3. FRP parameterization using the FFNNs**
Figure 1 illustrates the correlation skill and root-mean-squared error (RMSE) between
the observed FRP anomalies from 2001 to 2020 and the FRP anomalies estimated with
FFNNs and the FWI-based model. The correlation skill of the FFNNs exceeded 0.6
over southern China, northern India, southern South America, the eastern US, southern
Africa, western-central Russia, and maritime continents (Figure 1a). In contrast, the
correlation skill of the FWI-based model fell below 0.6, with southern China and central
Africa being the only exceptions (Figure 1b). Therefore, the FFNNs consistently
exhibited superior correlation skills compared to the FWI-based model over most of the
globe (Figure 1c). Notably, the improvement in the correlation skill of the FFNNs was
statistically significant at a 95% confidence level, as determined using the method
outlined by Zou (2007). This significance was particularly pronounced over East Asia,
the entirety of Russia, the eastern US, southern South America, and central Africa.
The RMSE of the FRP estimations tended to be higher over the regions with high
FRP climatology in both models (Laurent et al., 2019). A clear distinction in the RMSE
emerges upon comparing FFNNs and the FWI-based model; FFNNs demonstrate an
RMSE below 1.5 MW across most regions (Figure 1d), while the FWI-based model
predominantly registers RMSE values ranging between 1.5 and 1.8 MW (Figure 1e).
Consequently, the global depiction of RMSE differences reveals negative values,



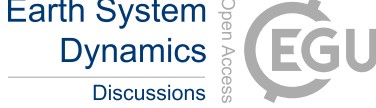

illustrating the consistent superiority of FFNNs over the FWI-based model (Figure 1f).
The systematic improvement in the accuracy of the estimated FRP using the
FFNNs was consistently robust when evaluating the skill evaluation exclusively for the
fire events (i.e., observed FRP > 0) (Supplementary Fig. S2) or when considering
monthly-averaged FRP anomalies (Supplementary Fig. S3); both daily and interannual
variations in estimated FRPs with FFNNs align more closely with the observed FRPs
across diverse regions than the corresponding outputs of the FWI-based model
(Supplementary Fig. S4). These findings highlight the superiority of FFNNs over the
FWI-based model in estimating fire events of varying temporal scales worldwide by
successfully exploring the relationship between the FRP and the meteorological
observations.
To identify the main factors that contributed to the superior accuracy of the
FFNNs, sensitivity experiments were conducted by fixing one of the meteorological
observations to the daily climatological values (Figure 2). The alteration in correlation
skill resulting from controlling each meteorological observation provides a measure of
the relative significance of the respective variable in FRP parameterization. The
correlation skill remained relatively unaffected when daily climatological values of
WS10m or T2m were considered (Supplementary Fig. S5 for the FFNNs, and
Supplementary Fig. S6 for the FWI-based model), indicating that the RH2m and PRCP
are the main factors influencing the accuracy of the FRP estimations in both models.
However, upon further examination, the degree of contribution from RH2m and
PRCP to the FRP estimations varied discernibly between FFNNs and the FWI-based
model. In the FFNNs, the correlation skill difference between the original estimation
and the estimation with the climatological RH2m was close to 0.5 over most of the
regions where the original FRP estimations exhibited high skill (Figure 2a). On the
other hand, substituting PRCP with its climatological value had a negligible impact on
the FFNN-based approach (Figure 2b). Therefore, RH2m was the dominant variable
influencing FRP estimations via the FFNNs method over most of the globe except for
a few regions (Figure 2c).
To support our findings, we adapted the layer-wise relevance propagation (LRP)
technique (Bach et al., 2015; Barns et al., 2020; Toms et al., 2020), which is widely
used for understanding the relevance of individual features or neurons in neural
networks. This analysis consistently validated the significance of RH2m as the most
sensitive factor influencing FRP estimation in FFNNs, with the contributions of other
meteorological parameters being comparatively minor (Supplementary Fig. S7).
Conversely, when employing the FWI-based model, the alteration in FRP
correlation skill is more pronounced upon substituting PRCP with its daily
climatological values. In regions such as southern China, northern India, southeastern
South America, and the eastern US, the correlation skill decrease is between 0.2 and
0.3 due to this substitution. In contrast, replacing RH2m with its climatology results in
correlation skill differences of less than 0.1 (Figure 2d and 2e). These findings
underscore the importance of PRCP as the meteorological variable with the greatest
influence on FRP estimation using the FWI (Figure 2f).
The dramatic disparity in the relative contributions of RH2m and PRCP between
the two models indicates that the factors that drive the predictive performance of the
two models were different. Therefore, the relationship between these two key
meteorological observations and the FRP estimations will be further explored in the
next section to gain insights into the factors that determine the superior performance of
the FFNN-based approach.

**4. Physical explanations of the superior performance of FFNNs**
To confirm that the superior performance of the FFNNs is associated with the
differences in the relationship between the RH2m and the estimated FRP between the
FFNNs and the FWI-based models, we selected grid points that satisfy the following
three conditions: (1) an FRP correlation skill improvement in FFNNs over FWI-based
models greater than 0.05, (2) RH2m as the most sensitive meteorological variable for
FRP estimation in FFNNs, and (3) PRCP as the most sensitive variable in the FWI-
based model. A total of 852 grid points were selected based on these criteria, which
accounts for approximately 25.1% of total land grid points and 49.7% of total grid
points whose correlation skill improvement in the FFNNs is greater than 0.05. The
selected grid points are located over southern China, Russia, central Africa, the eastern
US, and central-northern South America (Supplementary Fig. S8).
Figure 3 illustrates the averaged FRP for each RH2m bin with a 10% interval.
Our findings indicated that FRP exhibits a decrease when RH2m surpasses 30% (Figure
3a). Therefore, the difference in the FRP values in the higher RH2m bin from that in
the lower RH2m bin exhibited negative values (Figure 3b). This relationship reflects
the well-known impact of relative humidity on combustion, as oxygen availability is
constrained, resulting in reduced combustion rate and lowered FRP (Wooster et al.,



2005). Additionally, higher humidity can indicate the presence of moisture in the fuel,
such as plants or other vegetation, thereby impeding fire propagation and further
decreasing the FRP values.

Interestingly, in instances where RH2m falls below 30%, FRP tends to increase
with higher RH2m values. Although this proportional relationship between relative
humidity and fire activity is relatively uncommon, it can occur following extended
periods of drought or low humidity. For example, Abatzoglou and Kolden (2013)
reported that this phenomenon can arise when sudden moisture influx induces rapid
water uptake by vegetation, thus potentially intensifying fire activity.

The FFNNs accurately simulated the aforementioned nonlinear relationship
between the RH2m and the FRP (Figure 3c and 3d). In cases where RH2m < 30%, FRP
increases with rising RH2m; for RH2m > 30%, FRP diminishes as RH2m rises. The
consistency between the estimated and observed FRP values at each bin further
supports our previous results, demonstrating the successful application of FFNNs in
FRP parameterization.

In contrast, the FWI-based FRP estimations exhibit a linear inverse relationship
between the RH2m and the FRP. Specifically, FRP decreases continuously with
increasing RH2m (Figures 3e and 3f). This unrealistic representation, particularly in
dry regimes, demonstrates that the observed nonlinear RH2m-FRP relationship was not
faithfully captured in the FWI-based model. Furthermore, the FWI-based estimations
tended to overestimate FRP in low RH2m bins (i.e., RH2m < 30%) and underestimate
it in high RH2m bins (i.e., RH2m > 60%), which underscores the systematic biases in
the FRP estimations in the FWI-based model.

Next, we assessed the relationship between PRCP and the FRP values (Figure 4).
In both the observed FRP values and those estimated using FFNNs and FWI-based
models, PRCP tended to inhibit fire events, causing FRP values to decrease with rising
PRCP (Parks et al., 2014; Chen et al. 2014; Holden et al., 2018). In the observational
data (Figure 4a), FRP reaches its maximum at 1.9 MW within the lowest PRCP bin (i.e.,
PRCP < 0.1 mm/day), after which it sharply decreases to approximately 1 MW in the
subsequent bin (i.e., 0.1 mm/day < PRCP < 0.2 mm/day). Afterward, it experiences a
gradual decrease with increasing PRCP when PRCP is below 3 mm/day. However, for
PRCP values exceeding 3 mm/day, the extent to which FRP decreases with higher
PRCP becomes less pronounced, as higher precipitation does not proportionally reduce
ignition likelihood (Oliveras et al., 2014). This leads to sustained FRP values above a
certain threshold (i.e., 0.5 MW) for PRCP > 3 mm/day. The spatially averaged FRP
distribution in instances where PRCP > 3 mm/day maintains moderate values, ranging
from 1 to 2 MW over regions such as Mexico, Colombia, central South America, central
Africa, central Western Asia, Australia, and the maritime continent (Figure 4b).

FFNNs accurately simulated the observed relationship between the FRP and the

PRCP, with the estimated FRP in FFNNs exhibiting high values within the smallest
PRCP bins (approximately 1.75 MW), which decreased as PRCP increased when PRCP
was below 3 mm/day (Figure 4c). The spatial distribution of the averaged FRP for the
cases where PRCP > 3 mm/day was also similar to the observed values (Figure 4d).
Conversely, FRP estimation in the FWI-based model tended to be underestimated,
particularly in bins with higher PRCP (Figure 4e). For instance, bins with PRCP < 0.5
mm/day exhibited an underestimation of approximately 0.25 MW, whereas
underestimations of over 0.5 MW, and nearly 0 MW, were evident when PRCP > 3
mm/day. This suggests that the FWI-based model is more responsive to changes in
PRCP, resulting in a more pronounced FRP decrease with increasing PRCP. This is
further evidenced by the spatially averaged FRP distribution for PRCP > 3 mm/day,
which is almost negligible worldwide (Figure 4f).

This tendency aligns with the quadratic lines fitted to FRP values within each

PRCP bin; the quadratic coefficients for observations, FFNNs, and the FWI-based
model are 0.022, 0.023, and 0.036, respectively. The heightened FRP sensitivity to
PRCP changes in the FWI-based model contributes to the excessive influence of PRCP
on the FRP estimations, as shown in Figure 2f.

**5. Summary and Discussion**

In this study, we developed a parameterization method using FFNNs to estimate

global gridded FRP fields from meteorological variables. In the FFNNs, four daily
meteorological observations, namely 2 m temperature, 2 m specific humidity, wind
speed, and precipitation, were used as the input to predict the daily FRP output. The
cross-validated FRP parameterization results during 2001–2020 exhibited an improved
skill in estimating the observed FRP compared to the FWI-based linear regression
model. The improvement in the parameterization accuracy in terms of the correlation
skill and the RMSE was observed over most of the globe and was particularly
prominent over East Asia, Russia, the eastern US, southern South America, and central
Africa. This indicates that FFNNs can more effectively capture the nonlinear





relationship between meteorological observations and FRP compared to the commonly
employed fire index.

A series of sensitivity experiments were performed by replacing each variable

with the daily climatological values, and our findings demonstrated that the 2 m relative
humidity (RH2m) was the most critical variable influencing the outcomes of the FFNNs
over most of the globe. On the other hand, in the FWI-based model, PRCP plays a more
substantial role in FRP estimation. The observed nonlinear relationship between the
RH2m and the FRP is well simulated in the FFNNs; both the observation and the
FFNNs exhibited a negative relationship in the wet regime (i.e., RH2m > 30%),
whereas a positive relationship was observed in the dry regime (i.e., RH2m < 30%).
Likewise, FFNNs accurately simulated the observed impact of PRCP on FRP reduction.

In contrast, the FWI-based model simulated a linear negative relationship

between the FRP and the RH2m, which caused systematic errors in estimating the FRP,
particularly in the dry regime. Moreover, the FWI-based model exaggerates the degree
of FRP reduction with increasing PRCP, which contributes to the stronger contribution
of PRCP to the FRP estimations compared to those obtained with the FFNNs. This
discrepancy underscores the applicability of FFNNs in understanding the intricate
relationship between meteorological observations and FRP, offering insights for
refining the algorithm for global FWI calculations. While process-based fire models are
valuable for estimating fire activity changes due to greenhouse gas warming, their
performance is comparatively less robust compared to empirical models (Rabin et al.,
2015; Hantson et al., 2016). Therefore, FFNN parameterizations could enhance
process-based land surface models, yielding reliable fire activity predictions and
insights into their evolution under greenhouse gas warming scenarios.

Current FFNNs solely leverage meteorological observations for FRP

parameterization to ensure equitable comparison with the FWI-based model. However,
the incorporation of land surface observations such as soil moisture could optimize
FFNNs for simulating fire events more effectively. This provides an opportunity to
reduce the significant uncertainties in predicting fire events in parameterizing fires in
earth system models, ultimately mitigating potential losses from natural hazards.

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

Index (FWI) to improve the estimation of fire emissions from fire radiative power
(FRP)    observations, Atmospheric    Chem.    Phys., 18,    5359–5370,
https://doi.org/10.5194/acp-18-5359-2018, 2018.
Di Giuseppe, F., Pappenberger, F., Wetterhall, F., Krzeminski, B., Camia, A., Libertá,
G., and San Miguel, J.: The potential predictability of fire danger provided by
numerical weather prediction, J. Appl. Meteorol. Climatol., 55, 2469–2491,
https://doi.org/ 10.1175/JAMC-D-15-0297.1, 2016.
Duncan, B. N., Martin, R. V., Staudt, A. C., Yevich, R., and Logan, J. A.: Interannual
and seasonal variability of biomass burning emissions constrained by satellite
observations, J. Geophys. Res., 108, 4100, https://doi.org/10.1029/2002JD002378,
2003.
Flannigan, M. D., Logan, K. A., Amiro, B. D., Skinner, W. R., and Stocks, B. J.: Future
area burned in Canada, Clim. Change, 72, 1–16, https://doi.org/10.1007/s10584-
005-5935-y, 2005.
Giglio, L., Schroeder, W., and Justice, C. O.: The collection 6 MODIS active fire
detection algorithm and fire products. Remote Sensing of Environment, 178, 31–
41, https://doi.org/10.1016/j.rse.2016.02.054, 2016.
Grillakis, M., Voulgarakis, A., Rovithakis, A., Seiradakis, K. D., Koutroulis, A., Field,
R. D., Kasoar, M., Papadopoulos, A., and Lazaridis, M.: Climate drivers of global
wildfire    burned    area,    Environ.    Res.    Lett.,    17,    045021,
https://doi.org/10.1088/1748-9326/ac5fa1, 2022.





Hantson, S., Arneth, A., Harrison, S. P., Kelley, D. I., Prentice, I. C., Rabin, S. S.,
Archibald, S., Mouillot, F., Arnold, S. R., Artaxo, P., Bachelet, D., Ciais, P.,
Forrest, M., Friedlingstein, P., Hickler, T., Kaplan, J. O., Kloster, S., Knorr, W.,
Lasslop, G., Li, F., Mangeon, S., Melton, J. R., Meyn, A., Sitch, S., Spessa, A.,
van der Werf, G. R., Voulgarakis, A., and Yue, C.: The status and challenge of
global fire modelling, Biogeosciences, 13, 3359–3375, https://doi.org/10.5194/bg-
13-3359-2016, 2016.
Hersbach, H., Bell, B., Berrisford, P., Hirahara, S., Horányi, A., Muñoz-Sabater, J.,
Nicolas, J., Peubey, C., Radu, R., Schepers, D., Simmons, A., Soci, C., Abdalla,
S., Abellan, X., Balsamo, G., Bechtold, P., Biavati, G., Bidlot, J., Bonavita, M.,
De Chiara, G., Dahlgren, P., Dee, D., Diamantakis, M., Dragani, R., Flemming, J.,
Forbes, R., Fuentes, M., Geer, A., Haimberger, L., Healy, S., Hogan, R. J., Hóm,
E., Janisková, M., Keeley, S., Laloyaux, P., Lopez, P., Lupu, C., Radnoti, G., de
Rosnay, P., Rozum, I., Vamborg, F., Villaume, S., and Thépaut, J.-N.: The ERA5
Global Reanalysis, Q. J. Roy. Meteorol. Soc., 146, 1999–2049,
https://doi.org/10.1002/qj.3803, 2020.
Holden, Z. A., Swanson, A., Luce, C. H., Jolly, W. M., Maneta, M., Oyler, J. W.,
Warren, D. A., Parsons, R., and Affleck, D.: Decreasing fire season precipitation
increased recent western US forest wildfire activity, P. Natl. Acad. Sci. USA, 115,
201802316, https://doi.org/10.1073/pnas.1802316115, 2018.
Jones, M. W., Abatzoglou, J. T., Veraverbeke, S., Andela, N., Lasslop, G., Forkel, M.,
Smith, A. J. P., Burton, C., Betts, R. A., van der Werf, G. R., Sitch, S., Canadell,
409        J. G., Santín, C., Kolden, C., Doerr, S. H., and Le Quéré, C.: Global and regional
trends and drivers of fire under climate change, Rev. Geophys., 60,
e2020RG000726, https://doi.org/10.1029/2020RG000726, 2022.
Kim, J. S., Jeong, S. J., Kug, J. S., and Williams, M.: Role of local air-sea interaction
in fire activity over equatorial Asia, Geophys. Res. Lett., 46, 14789–14797,
https://doi.org/10.1029/2019GL085943, 2019.
Kim, J. S., Kug, J. S., Jeong, S. J., Park, H., and Schaepman-Strub, G.: Extensive fires
in southeastern Siberian permafrost linked to preceding Arctic Oscillation, Sci.
Adv., 6, eaax3308, https://doi.org/10.1126/sciadv.aax3308, 2020.
Laurent, P., Mouillot, F., Moreno, M. V., Yue, C., and Ciais, P.: Varying relationships
between fire radiative power and fire size at a global scale, Biogeosciences, 16,
275–288, https://doi.org/10.5194/bg-16-275-2019, 2019.
Le Page, Y., van der Werf, G. R., Morton, D. C., and Pereira, J. M. C.: Modeling fire-
driven deforestation potential in Amazonia under current and projected climate
conditions, J. Geophys. Res.-biogeosciences, 115, G03012,
https://doi.org/10.1029/2009JG001190, 2010.
Macias Fauria, M., Michaletz, S. T., and Johnson, E. A.: Predicting climate change
effects on wildfires requires linking processes across scales, Wiley
Interdisciplinary Reviews: Climate Change, 2, 99–112,
https://doi.org/10.1002/wcc.92, 2011.
Oliveras, I., Anderson, L. O., and Malhi, Y.: Application of remote sensing to
understanding fire regimes and biomass burning emissions of the tropical Andes,
Global Biogeochem. Cy., 28, 480– 496, https://doi.org/10.1002/2013GB004664,
2014.
Haas, O., Prentice, I. C., and Harrison, S. P.: Global environmental controls on wildfire
burnt area, size, and intensity, Environ. Res. Lett., 17, 065004,
https://doi.org/10.1088/1748-9326/ac6a69, 2022.



Parks, S. A., Parisien, M.-A., Miller, C., and Dobrowski, S. Z.: Fire Activity and
Severity in the Western US Vary along Proxy Gradients Representing Fuel
Amount and Fuel Moisture, PLoS ONE, 9, e99699,
https://doi.org/10.1371/journal.pone.0099699, 2014.
Rabin, S. S., Melton, J. R., Lasslop, G., Bachelet, D., Forrest, M., Hantson, S., Kaplan,
J. O., Li, F., Mangeon, S., Ward, D. S., Yue, C., Arora, V. K., Hickler, T., Kloster,
S., Knorr, W., Nieradzik, L., Spessa, A., Folberth, G. A., Sheehan, T., Voulgarakis,
A., Kelley, D. I., Prentice, I. C., Sitch, S., Harrison, S., and Arneth, A.: The Fire
Modeling Intercomparison Project (FireMIP), phase 1: experimental and
analytical protocols with detailed model descriptions, Geosci. Model Dev., 10,
1175–1197, https://doi.org/10.5194/gmd-10-1175-2017, 2017.
Rogers, B. M., Soja, A. J., Goulden, M. L., and Randerson, J. T.: Influence of tree
species on continental differences in boreal fires and climate feedbacks, Nat.
Geosci., 8, 228–234, https://doi.org/10.1038/ngeo2352, 2015.
Toms, B. A., Barnes, E. A., and Ebert-Uphoff, I.: Physically interpretable neural
networks for the geosciences: Applications to Earth system variability, J. Adv.
Model. Earth Syst., 12, e2019MS002002, https://doi.org/10.1029/2019ms002002,
2020.
Van Wagner, C. E.: Structure of the Canadian forest fire weather index, Can. For. Serv.
Publ., 1333, 44 pp., 1974.
Van Wagner, C. E.: Development and structure of the Canadian forest fire weather
index system, Canadian Forestry Service, Headquarters, Ottawa, Canada, Forestry
Technical Report, vol. 35, 35 pp., 1987.
Vitolo, C., Di Giuseppe, F., Krzeminski, B., and San-Miguel-Ayanz, J.: A 1980–2018
global fire danger re-analysis dataset for the Canadian Fire Weather Indices, Scient.
Data, 6, 190032, https://doi.org/10.1038/sdata.2019.32, 2019.
Wooster, M. J., Roberts, G., Perry, G., and Kaufman, Y.: Retrieval of biomass
combustion rates and totals from fire radiative power observations: FRP derivation
and calibration relationships between biomass consumption and fire radiative
energy release, J. Geophys. Res., 110, D24311,
https://doi.org/10.1029/2005JD006318, 2005.
Zou, G.: Toward using confidence intervals to compare correlations, Psychol. Methods,
12, 399–413, https://doi.org/10.1037/1082-989X.12.4.399, 2007.




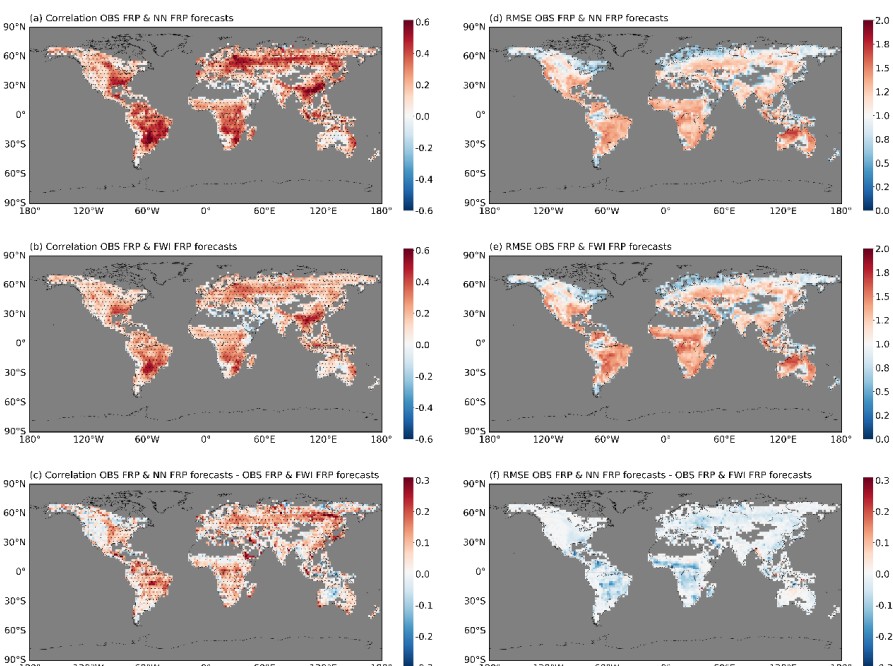


Figure 1. Correlation skill between the observed daily FRP and the estimated FRP values in (a) the FFNNs or (b) FWI-based linear regression model during 2001–2020. (c) Difference in the correlation skill in the FFNNs from that in the FWI-base model. RMSEs between the observed daily FRP and the estimated FRP values in (d) the FFNNs, or (e) FWI-based linear regression model during 2001–2020. (f) Difference in the RMSE in the FFNNs from that in the FWI-base model. The dots in panels (a) and (b) denote the grid points where the correlation skill exceeds a 95% confidence level based on the t-test; those in panel (c) denote the area whose correlation skill difference is above a 95% confidence level calculated as described by Zou (2007).





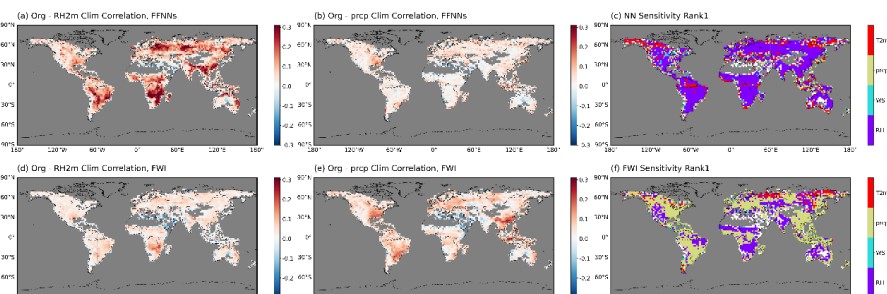


Figure 2. Difference in the correlation skill of the original FRP estimation in the
FFNNs from that by prescribing (a) the RH2m or (b) the PRCP as the daily
climatological values. (c) Spatial distribution of the meteorological variable where the
decrease in correlation is largest by prescribing the climatological value. Panels (d), (e),
(f) are the same as (a), (b), and (c) but for the FWI-based model. In panels (c) and (f),
2 m air temperature, PRCP, 10 m wind speed, and RH2m are indicated in red, yellow,
green, and purple, respectively.




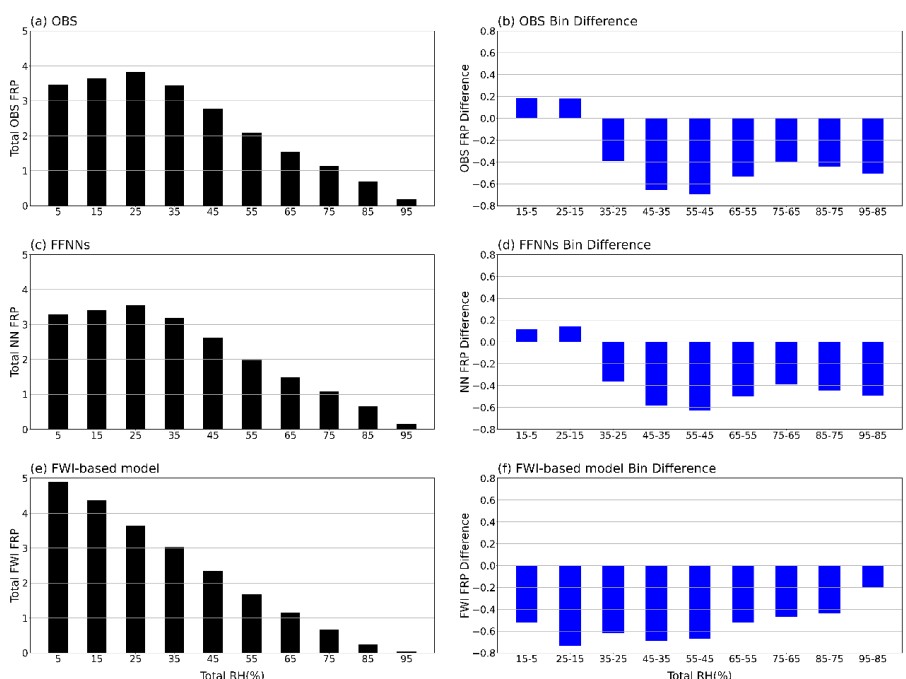


Figure 3. Case-averaged FRP with respect to the RH2m with a 10% interval in (a)
the observations, (c) FFNNs, and (e) FWI-based model. The figures illustrate the
difference in the case-averaged FRP at the upper bin from the lower bin in (b) the
observations, (d) FFNNs, and (f) FWI-based model. The selected areas for the
calculation are shown in Supplementary Fig. S8.




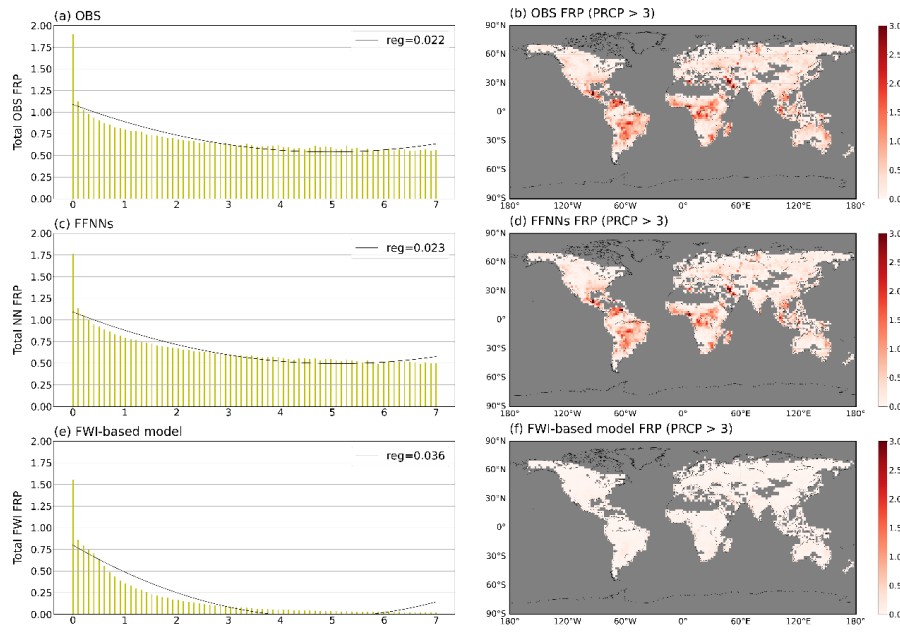


Figure 4. Case-averaged FRP with respect to the PRCP with 0.1 mm/day interval in
(a) the observations, (c) FFNNs, and (e) FWI-based model. The figures illustrate the
spatial distribution of the case-averaged FRP when the PRCP > 3 mm/day in (b) the
observations, (d) FFNNs, and (f) the FWI-based model.