# Peer review of "Yoo-Geun Ham1, Seung-Ho Nam1, and Jin-Soo Kim2"

_Earth System Dynamics, 2023_

## Referee Comment (RC1)

**Reviewer comments for esd-2023-26**

**Title: Regionally optimized fire parameterizations using feed-forward neural networks**

Authors: Yoo-Geun Ham, Seon-Ho Nam, and Jin-Soo Kim

In this manuscript, the authors have proposed a neural network based method to simulate Fire Radiative Power (FRP). The inputs to this network are 4 meteorological parameters. This new method to estimate FRP can be useful to scientists involved in understanding the fire intensity and weather/climate relationships, especially with less access to various fire-related datasets like ignition etc. However, there are some serious issues in the methods and interpretation of the results obtained. I recommend addressing the methodological issues and rethinking the interpretations of the results before proceeding with publication. Also, the composition of the paper needs to be changed significantly. I have segregated my comments into three sections, major, minor and language-related. Following are the points in the order of seriousness.

**Major comments:**

1. **Methodological improvement with FFNN**

I found several instances which do not strongly support the opinion of the authors, i.e., FFNN-based FRP estimates are indeed better than the FWI-based ones.

A. The whole argument is based on the comparison of FFNN-based FRP estimates with the FWI-based linear regression model. Why linear regression? The relationship between FWI and FRP is not at all linear. The correct approach would be to use various non-linear regression techniques, take the best of them and then compare them with the FFNNs. Only then can we compare the predictive abilities between FWI-based and FFNN-based methods. I expect significant changes in results and conclusion if instead of the linear method some other non-linear method is used.

B. Though FWI was originally developed for Canada, a large number of studies have used the system successfully to ascertain fire weather. It is true that the equations do not take into account some factors like species distribution etc. (lines 57-59). Even FFNN or in fact, any machine-learning-based model also does not take these into account.

C. Inferences in lines 60-62 such as which variable contributes most to Arctic or Amazon fire activity can be determined by FWI-based studies too.

D. Lines 156-158 and Figure 1.b:  Even if this approach of linear regression is considered, from the figure it seems that the FWI-based model is statistically significant at all points as FFNNs. How is that an improvement?

E. Line 249: Since a linear regression method was used to map FWI into FRP, of course, the FRP estimates would not be able to capture the non-linear RH2m-FRP and PRCP-FRP relationships. The argument in section 4 needs to be rewritten after considering some non-linear regression of FWI and FRP.

F. Methods like FFNNs need extensive validation. Though it is a common practice to train and validate ML models using the same datasets, if additional validation can be done with any station data or ground observations, then the FRP estimates will be more reliable.

2. **Incomplete information in the methods**

A. Methods should include all the calculations and steps taken for complete analysis leading to figures and also stating the reasons for conducting that particular analysis. The sensitivity experiments are not explained elaborately. A flowchart or graphical representation of these steps or at least a detailed explanation would be appropriate. Otherwise, it is very cumbersome for readers to understand the inferences discussed in the subsequent sections.

B. Input variables for the FFNN model as well as for computing FWI are discussed in lines 95-96. Are these taken at 12 Noon local time? Or daily averages? For precipitation, was it the 24-hour sum or average? Ideally, 12 Noon UTC values or daily maximum temperature, minimum relative humidity, average wind speed and 24-hour accumulated precipitation should be used. Please specify in the manuscript.

C. In lines 118-121, some techniques related to the neural networks are just mentioned. Please explain the terms or provide citations which explain the terms like dropout rate, batch normalization, and ReLU function for readers not familiar with these. Also, what does a dropout rate of 0.2 physically signify? Please explain these in the text.

3. **Miscellaneous**

A. An anomalous behaviour of increasing FRP with increasing RH when humidity is less than 30% has been reported in section 4 (lines 237-241). This is supported by an argument reported by Abatzoglu and Kolden (2013). However, I could not find any such observation in the cited article. Can the authors clarify exactly where in the paper this is mentioned? Also, fire activity usually translates to fire frequency. FRP gives us more information about fire intensity. It is unusual that fire intensity at 20% RH will be lesser than at 30% RH.

B. In supplementary figure S5, the result of precipitation and temperature values are almost similar. Line 182 in the main text contradicts this.

C. In lines 167-170. Supplementary figures S2 and S3 have been discussed. These figures refer to Fig.2 in the main article which has not been discussed yet. Such issues in the chronology of the paper disrupt the course of reading. The whole of the result sections need to be reorganised.

D. Lines 186-194: Inferences related to FFNNs are discussed here, however, the inferences from the FWI-based method (Figure 2d-f) are in lines 201-208. These inferences related to the same figure should be kept together. Also, between Fig 2c and Fig 2f, how do we conclude which is more correct?

E. Line 195: Explain the physical interpretation of the LRP method for unfamiliar readers. How is the result obtained from this analysis different from the sensitivity study with FFNNs? It is still unclear which method gives a more accurate estimate of the major factors influencing FRP. Also, explain supplementary figure S7 in the supplementary or main text.

F. Line 221: Why was this particular 0.05 number chosen? The sentence in lines 223-225 is unclear. Please rephrase. Also, figure S8 can be moved to the main article.

G. Line 233: What is the relevance of this reference here?

H. Line 257 onwards: Is the precipitation considered here the daily average or the daily sum? The ideal approach would be to use daily sum. Clarify.

I. Line 285-289: Is quadratic the best fit? Also, please explain why the heightened sensitivity of FRP to precip is incorrect. That very well might be the case.

J. ERA5 has been used in this study. What will be the scenario if we use some other weather data/ observation? How sensitive is the FFNN model to the kind of dataset used?

K. In line 136, it is said that the entire period is divided into three-year periods. But if the test period is 1st Jan 2001- 31st Dec 2004, it is actually a four-year period. Such confusion should be removed.

L. Supplementary figure S4: Why these particular stations and years? This is nowhere mentioned in the methods/ results/ supplementary text. Please elaborate.

M. In Line 300, FRP behaviour or rather correlation of FFNN estimated FRP and observations over certain regions are discussed. However, why certain regions show high/low correlation has not been discussed anywhere.

4. **Regarding figures:**

A. What does except 0 mean in supplementary figure S2? Mention in the text/ supplementary the methodology of how these figures were obtained, their necessity and inferences.

B. A reader has to constantly toggle between main article figures and supplementary figures. The order in which figures are discussed is random

**Minor comments:**

1. Lines 26-27: This sentence requires rephrasing. The FFNNs captured the 'relationship' accurately. Correlation is a method by which we can ascertain this relationship. Also, what do the authors mean by 'as well as precipitation'? Is precipitation well correlated or not? Please clarify.

2. Line 28: Ideally, we expect an inverse relationship between FRP and precipitation. How is this 'excessive' relationship a concern in this context?

3. Line 34: What kind of fires? Wildfires or agricultural fires? Not all fires cause ecological and socio-economic impacts. Please specify.

4. Lines 36-37: I see no relevance of this statement here, as the authors have not discussed the ignition factor anywhere in the manuscript.

5. Line 49: Moisture codes provide no exclusive information about any deceased organic matter.

6. Line 63: It is unclear what the authors are trying to convey here. Also, how is it relevant to the rest of the paragraph?

7. Line 65: How can we 'calibrate' sensitivity? Consider changing it to 'estimate/ calculate'.

8. Line 70: I guess there is a typo. It should be 'fuel' moisture code instead of fire.

9. Line 116: consider changing 'responsible for' with 'representing'

10. Line 127: there is a typo in the equation. It should be $y_i$ instead of $y_1$.

11. Line 132: Is the title 'Experimental design' 'for section 2.3 suitable? It is rather an explanation of the cross-validation strategy only.

12. Line 143-145: Rephrase the sentence. Anomalies were compared and assessed for accuracy.

13. Line 161: Please provide a map of FRP climatology in the main or supplementary article. What does the citation here convey?

14. Line 196: To support which findings? Please specify.

15. Line 213, 217: The ideal course for this kind of study should be first to identify which is a better method, gather sufficient evidence and then put forth the result as to which is a better method. Also, try to avoid phrases like 'confirm the superior performance'.

16. Line 216: What does 'Physical' explanation mean? Consider changing the title of this section.

**Language suggestions:**

1. The tense of the entire article needs to be rechecked. For example, in line 228 the authors have used present tense, but in line 229, 'indicated' is past tense.

2. Avoid using unnecessary adjectives and adverbs. For example, in line 21, 'marked' enhancement is obsolete. Similarly, in line 209, 'dramatic' disparity is unnecessary. Consider removing all such adjectives and adverbs unless absolutely necessary as they do not add any additional merit to the manuscript.

3. Paragraphing should be rechecked. Ensure that all sentences within a paragraph are related to the main idea expressed in it. For example, the information in the paragraph starting at line 313 is related to the previous paragraph. In such a case a paragraph break is not required. The authors should review the whole article for appropriate paragraphing.

4. Line 44: 'holds a prominent status' is not suitable here. There are other indices also which are used in various countries for operational purposes. FWI is one of the popular ones. Please rephrase the sentence.

5. Line 175: Consider changing 'exploring' to 'simulating/ calculating/ estimating'. The relationship is known and already established. The models are just simulating them.

6. Line 179: 'alteration' is not suitable here

7. Line 232: A sentence break is required before 'as oxygen…'

---

## Author Comment (AC1)

**Reviewer comments for esd-2023-26**

**Title: Regionally optimized fire parameterizations using feed-forward neural networks**

Authors: Yoo-Geun Ham, Seon-Ho Nam, and Jin-Soo Kim

In this manuscript, the authors have proposed a neural network based method to simulate Fire Radiative Power (FRP). The inputs to this network are 4 meteorological parameters. This new method to estimate FRP can be useful to scientists involved in understanding the fire intensity and weather/climate relationships, especially with less access to various fire-related datasets like ignition etc. However, there are some serious issues in the methods and interpretation of the results obtained. I recommend addressing the methodological issues and rethinking the interpretations of the results before proceeding with publication. Also, the composition of the paper needs to be changed significantly. I have segregated my comments into three sections, major, minor and language-related. Following are the points in the order of seriousness.

**Major comments:**

**1. Methodological improvement with FFNN**

I found several instances which do not strongly support the opinion of the authors, i.e., FFNN-based FRP estimates are indeed better than the FWI-based ones.

A. The whole argument is based on the comparison of FFNN-based FRP estimates with the FWI-based linear regression model. Why linear regression? The relationship between FWI and FRP is not at all linear. The correct approach would be to use various non-linear regression techniques, take the best of them and then compare them with the FFNNs. Only then can we compare the predictive abilities between FWI-based and FFNN-based methods. I expect significant changes in results and conclusion if instead of the linear method some other non-linear method is used.

: We fully understand the reviewer's concern. Our reference method to be compared to FFNN is FWI-based forecasts, and the linear regression is just to match the variability between FWI index and the FRP. In other words, we compared the parameterization quality seeking the nonlinearity between the meteorological variable and the FRP in the widely-used meteorology-based fire intensity estimation algorithm (i.e., FWI algorithm) to that seeking the nonlinearity using the neural network weights and the nonlinear activations.

To avoid the confusion, we modified the term 'FWI-based linear regression model' to 'FWI-based model' throughout the revised manuscript, and the brief description about the FWI-based model is also modified in the revised manuscript as follows.

Line 128-136 : "A FRP-estimation model based on the FWI was established as a baseline. The FWI is obtained from the daily averages of T2m, RH2m, WS10m, and PRCP, and .... To match the systematic amplitude differences between the FWI and FRP using the different units, a linear regression coefficient of the FRP with respect to the FWI, which was separately calculated for each grid point, is multiplied to produce the FWI-based model. Therefore, the nonlinearity between the meteorological variable and the FRP in the baseline model is purely originated from the procedure to derive the FWI."

B. Though FWI was originally developed for Canada, a large number of studies have used the system successfully to ascertain fire weather. It is true that the equations do not take into account some factors like species distribution etc. (lines 57-59). Even FFNN or in fact, any machine-learning-based model also does not take these into account.

: Thank you for point this out. Both FFNN and FWI cannot take into account the species distribution. The corresponding sentence is modified as follows "However, regional fire dynamics vary significantly depending on its unique climatological states (Flannigan et al., 2005, Kim et al., 2019)."

C. Inferences in lines 60-62 such as which variable contributes most to Arctic or Amazon fire activity can be determined by FWI-based studies too.

: We tried to argue that the regional difference between Amazon and Arctic region would not be fully considered as the algorithm for the FWI is purely based on the data in Canada. We modified the

sentence as follows.

Line 70-78 : ", however, its regional difference would not be fully considered as the FWI is purely derived by the data in Canada, while the relationship between the meteorological variables and the fire activity varies significant from regions to regions."

D. Lines 156-158 and Figure 1.b: Even if this approach of linear regression is considered, from the figure it seems that the FWI-based model is statistically significant at all points as FFNNs. How is that an improvement?

: The significant test for the difference between FWI model and NN model is already performed in original Figure 1c. For the significant of the difference, we followed the method in Zou (2007) as follows. First, Fisher's z transformation of correlation coefficient r for each forecasts is calculated $z = \frac{1}{2}\ln\left(\frac{1+r}{1-r}\right) = artanh(r)$. If (X, Y) has a bivariate normal distribution with correlation ρ and the pairs (Xi, Yi) are independent and identically distributed, then z is approximately normally distributed with mean of $\frac{1}{2}\ln\left(\frac{1+\rho}{1-\rho}\right)$, and standard deviation of $\frac{1}{\sqrt{N-3}}$, where N is the sample size, and ρ is the true correlation coefficient. Then, the difference in the standardized correlation, i.e., $\frac{1}{2}\ln\left(\frac{1+\rho1}{1-\rho1}\right)/\frac{1}{\sqrt{N1-3}} - \frac{1}{2}\ln\left(\frac{1+\rho2}{1-\rho2}\right)/\frac{1}{\sqrt{N2-3}}$, is over 0.05 (i.e., 95% confidence level) were marked.

E. Line 249: Since a linear regression method was used to map FWI into FRP, of course, the FRP estimates would not be able to capture the non-linear RH2m-FRP and PRCP-FRP relationships. The argument in section 4 needs to be rewritten after considering some non-linear regression of FWI and FRP.

: FWI can consider the nonlinear RH2m-FRP, or PRCP-FRP relationship as the procedure to derive the FWI from the meteorological variables are nonlinear (https://www.nwcg.gov/publications/pms437/cffdrs/fire-weather-index-system). For example, As shown in Figure 3, and 4 of the original manuscript (also shown in Figure A-1, and A-2 in this response letter), relationship between the RH2m and FWI, and PRCP and FWI is not linear, respectively; the differences between the bin is not a constant. As our main goal to compare the FWI model as a reference is that the simulation of the nonlinearity between the meteorological variables and the FRP in the FFNN is realistic compared to that in the FWI, we did not do any further corrections in FWI-based forecasts by applying non-linear regression of FWI and FRP.

[Figure]

Figure A-1. Case-averaged FRP with respect to the RH2m with a 10% interval in the FWI-based model (left), and the differences in the case-averaged FRP at the upper bin from the lower bin in FWI-based model (right).

[Figure]

Figure A-2. Case-averaged FRP with respect to the PRCP with 0.1 mm/day interval in the FWI-based model

F. Methods like FFNNs need extensive validation. Though it is a common practice to train and validate ML models using the same datasets, if additional validation can be done with any station data or ground observations, then the FRP estimates will be more reliable.
: Thank you for the constructive suggestion. However, it is extremely hard to obtain the daily ground observations for the enough period to validate the fire activity. We note the limitation of our study by not validating our forecasts with ground observations in the revised manuscript as follows.

Line 179-186 : "We note that evaluating the skill of FFNN against FRP data may lead to an overestimation of its estimation abilities, given that the FFNN is trained using same type of data. Regrettably, the absence of ground-based observations on fire activity/intensity for the enough period deprives us of the opportunity to cross-reference FFNN-based FRP estimations with independent observations."

2. **Incomplete information in the methods**

A. Methods should include all the calculations and steps taken for complete analysis leading to figures and also stating the reasons for conducting that particular analysis. The sensitivity experiments are not explained elaborately. A flowchart or graphical representation of these steps or at least a detailed explanation would be appropriate. Otherwise, it is very cumbersome for readers to understand the inferences discussed in the subsequent sections.

: Sorry for the inconvenience. As a reviewer mentioned, we added a detailed procedure about the sensitivity experiments for Figure 2, and layer-wise relevance propagation (LRP) for main Fig. 5 as follows.

For sensitivity experiments in Figure 2, Line 231-237 : "… in the RH2m Clim experiment, the prescribed values of RH2m as an input of the FFNN is the daily climatology during the whole period (i.e., 2001-2020), therefore, its year-to-year variations in the RH2m is removed. Then, the correlation skill difference between the control simulation, that prescribes all input values at the corresponding date, and the RH2m Clim experiment is calculated to assess the importance of the RH2m in FRP parameterization."

For LRP in Supp. Fig. 7, Line 261-274 : "It provides a so-called relevance score, which linearly decompose the importance of each input variables as follows by propagating the output value backward toward the input variables using a chain rule.

$$f(RH2m, PRCP, T2m, WS) = R_{RH2m} + R_{PRCP} + R_{T2m} + R_{WS}$$

where $f$ is a nonlinear model (i.e., FFNNs) to derive the FRP, and $R_{RH2m}$, $R_{PRCP}$, $R_{T2m}$, $R_{WS}$ is a relevance score of RH2m, PRCP, T2m, and WS10m, respectively. The relative importance of any particular variable to the estimated FRP can be quantified by calculating the degree of the similarity between the relevance scores. For this purpose, we obtained the relevance score of each variable for each day during the whole testing period (i.e., 2001-2020), and calculated the correlation with the estimated FRP in the FFNNs."

B. Input variables for the FFNN model as well as for computing FWI are discussed in lines 95-96. Are these taken at 12 Noon local time? Or daily averages? For precipitation, was it the 24-hour sum or

average? Ideally, 12 Noon UTC values or daily maximum temperature, minimum relative humidity, average wind speed and 24-hour accumulated precipitation should be used. Please specify in the manuscript.
: Sorry for missing an important information. It is 24-hour averaged value. This temporal treatment in the input variable is identical to the input for the FWI. We noted this information in the revised manuscript.

C. In lines 118-121, some techniques related to the neural networks are just mentioned. Please explain the terms or provide citations which explain the terms like dropout rate, batch normalization, and ReLU function for readers not familiar with these. Also, what does a dropout rate of 0.2 physically signify? Please explain these in the text.
: Sorry for the possible inconvenience. We added a short description of each term with references in the revised manuscript.

3. **Miscellaneous**
A. An anomalous behaviour of increasing FRP with increasing RH when humidity is less than 30% has been reported in section 4 (lines 237-241). This is supported by an argument reported by Abatzoglu and Kolden (2013). However, I could not find any such observation in the cited article. Can the authors clarify exactly where in the paper this is mentioned? Also, fire activity usually translates to fire frequency. FRP gives us more information about fire intensity. It is unusual that fire intensity at 20% RH will be lesser than at 30% RH.
: Sorry for causing the inconvenience. In Abatzoglou and Kolden (2013), even though they showed that the positive correlation between the soil moisture and the burned area in non-forested regions (compared their figure 4 to figure 3), they did not discuss its mechanism. About this positive relationship, Xystrakis et al. (2014) discussed that the increased wetness is associated with the build-up of the fuel during the drying season, which eventually contribute to increase the burned area. This is consistent with our results to some extent that the increased FRP with the increased relative humidity occurs in the low relative humidity regime, which might imply that the increased relative humidity contributes to increase the fire activity (i.e., FRP) by increasing the fuels to burn. We added the aforementioned discussions in Line 308-314 of the revised manuscript.

B. In supplementary figure S5, the result of precipitation and temperature values are almost similar. Line 182 in the main text contradicts this.
: Sorry for causing the confusion. This sentence is modified as follows, "It clearly indicates that the RH2m are the main factors influencing the accuracy of the FRP estimations in the FFNNs.".

C. In lines 167-170. Supplementary figures S2 and S3 have been discussed. These figures refer to Fig.2 in the main article which has not been discussed yet. Such issues in the chronology of the paper disrupt the course of reading. The whole of the result sections need to be reorganised.
: We looked through the paragraph describing Supp Fig. S2 and S3, but cannot find any statements mentioning Figure 2. Supp. Fig. S2 and S3 is the correlation skill after managing the data, which is the follow-up analysis of Figure 1, therefore, we think the results section is organized in orders.

D. Lines 186-194: Inferences related to FFNNs are discussed here, however, the inferences from the FWI-based method (Figure 2d-f) are in lines 201-208. These inferences related to the same figure should be kept together. Also, between Fig 2c and Fig 2f, how do we conclude which is more correct?
: Sorry for the inconvenience. We modified the orders of the paragraph as a reviewer suggested.

E. Line 195: Explain the physical interpretation of the LRP method for unfamiliar readers. How is the result obtained from this analysis different from the sensitivity study with FFNNs? It is still unclear which method gives a more accurate estimate of the major factors influencing FRP. Also, explain

supplementary figure S7 in the supplementary or main text.

: We added a description of the LRP method with a physical explanation in Line 261-274. It is hard to say which method between the LRP and the sensitivity experiment by replacing an input variable is more accurate to estimate the relative importance, and that is why we introduced both methods in the article. The LRP method provides an output so-called 'relevance score' for each variable, and, main Fig. 5 is the correlation skill between the estimated FRP and the relevance score. This information is also added in the revised manuscript.

F. Line 221: Why was this particular 0.05 number chosen? The sentence in lines 223-225 is unclear. Please rephrase. Also, figure S8 can be moved to the main article.

: We tested a threshold of 0.05 and 0.1 of correlation improvement, and found that the general conclusion is still obvious with different threshold (Figure A-3). The final threshold is chosen as 0.05 as the large number of selected grid points would provide a rigorous result. We noted this point in Line 294-296 of the revised manuscript.

Sentences in lines 287-289 is modified as follows "… (2) RH2m is the most sensitive variable for FRP estimation in FFNNs (green color in Fig. 2c), and (3) PRCP is the most sensitive variable in the FWI-based model (blue color in Fig. 2f)."

Also, we included Figure S8 in the main figure as a reviewer suggested.

[Figure]

Figure A-3. Same is main Fig. 6, but for the threshold of 0.1 for the correlation skill improvement.

G. Line 233: What is the relevance of this reference here?

: Sorry for the confusion. We changed the references to clearly state the relationship between the relative humidity and the wildfire as follows.

Papagiannaki, K., Giannaros, T. M., Lykoudis, S., Kotroni, V., & Lagouvardos, K. (2020). Weather-related thresholds for wildfire danger in a Mediterranean region: The case of Greece. Agricultural and Forest Meteorology, 291, 108076.

Ying, L., Cheng, H., Shen, Z., Guan, P., Luo, C., & Peng, X. (2021). Relative humidity and agricultural activities dominate wildfire ignitions in Yunnan, Southwest China: Patterns, thresholds, and

implications. Agricultural and Forest Meteorology, 307, 108540.

H. Line 257 onwards: Is the precipitation considered here the daily average or the daily sum? The ideal approach would be to use daily sum. Clarify.
: We utilized the daily averages for the precipitation. We noted this point in Section 2.1.2. As an input of the statistical model, a difference between daily average and daily sum should not affect to the final results, as a simple multiplication of the constant value converts daily averages into daily summation.

I. Line 285-289: Is quadratic the best fit? Also, please explain why the heightened sensitivity of FRP to precip is incorrect. That very well might be the case.
: In our view, the quadratic fit works relatively well for the bin ranging from 0.1 mm/day to 5 mm/day. It is to roughly quantify the changes in the observed or estimated FRP to the changes in the precipitation. The heightened sensitivity in the FWI-based model is caused by the abrupt drop of the estimated FRP to the increase of the precipitation amount from 0 to 3 mm/day as shown in main Fig. 7e. On the other hand, degree of the changes in the FRP in the observation or FFNNs according to the changes in the precipitation is systematical weaker than that in the FWI-based model. The discussion about the heightened sensitivity of FRP to precipitation in the FWI-based model is elaborated in Line 356-365 as follows

Line 356-365 : "As a result, the regression coefficient between the FRP estimation and the PRCP is systematically greater in the FWI-based model. For observations, the quadratic coefficient is 0.022 MW/(mm/day)$^2$ (black in Figure 4a), and that for the FFNNs 0.023 MW/(mm/day)$^2$ (black in Figure 4c), denoting similar amplitude. On the other hand, the FWI-based model is 0.036 MW/(mm/day)$^2$, which is almost twice to that of the others (black in Figure 4e). This suggests that the FWI-based model is more responsive to changes in PRCP, resulting in a more pronounced FRP decrease with increasing PRCP. This excessive sensitivity in the estimated FRP to PRCP changes can contribute to the excessive influence of PRCP on the FRP estimations in the FWI-based model, as shown in Figure 4f."

J. ERA5 has been used in this study. What will be the scenario if we use some other weather data/ observation? How sensitive is the FFNN model to the kind of dataset used?
: It might be worthwhile to check as a reviewer asked, however, we did not test with different dataset (e.g., MERRA2, NCEP), as 1) ERA5 is one of widely used reanalysis product, and 2) it is important to compare FFNNs and FWI by utilizing same dataset.

K. In line 136, it is said that the entire period is divided into three-year periods. But if the test period is 1st Jan 2001- 31st Dec 2004, it is actually a four-year period. Such confusion should be removed.
: Sorry for the mistake. It is corrected to 'four-year periods' as a reviewer pointed out.

L. Supplementary figure S4: Why these particular stations and years? This is nowhere mentioned in the methods/ results/ supplementary text. Please elaborate.
: We picked the stations and years to check the simulation quality of the major wildfire events.
(a,b) 2019 Amazon wildfire : https://en.wikipedia.org/wiki/2019_Amazon_rainforest_wildfires
(c,d) 2016 Congo wildfire : https://www.mdpi.com/2072-4292/8/12/986
(e,f) 2003 Siberian wildfire : https://www.tandfonline.com/doi/pdf/10.1080/01431160802541549
(g,h) 2007 Southern China wildfire : https://link.springer.com/article/10.1007/s13753-017-0129-6
We added this information in Line 216-228 of the revised manuscript.

M. In Line 300, FRP behaviour or rather correlation of FFNN estimated FRP and observations over certain regions are discussed. However, why certain regions show high/low correlation has not been discussed anywhere.
: The mechanism of the improvement is given in the subsequent paragraph; the sensitivity

experiments with the examination of the relationship between FRP and RH2m, or PRCP in Figure 3 and 4 is to understand the improvement of the FFNNs compared to the FWI-based model. That is why we included grid points whose correlation skill improvement in the FFNNs is greater than a threshold value of 0.05 for plotting Figure 3 and 4. As this would be good to clearly refer that the sensitivity experiments is to understand the improvement of the FFNNs, we modified the sentence in Line 373 as follows.

Line 379 : "To identify the mechanism of the skill improvement in the FFNNs, a series of sensitivity experiments were performed …"

**4. Regarding figures:**

**A. What does except 0 mean in supplementary figure S2? Mention in the text/ supplementary the methodology of how these figures were obtained, their necessity and inferences.**

: Sorry for the lack of explanations. It is to evaluate the forecast skill only with the fire events. In some case of many non-wildfire days (i.e., FRP = 0), the skill is quite high even though the model always predicts a constant value (i.e., 0). We added the brief meaning of FRP > 0 in Line 210-211.

**B. A reader has to constantly toggle between main article figures and supplementary figures. The order in which figures are discussed is random.**

: Sorry for the confusion. We double-checked that the figures are discussed in order throughout the revised manuscript.

**Minor comments:**

**1. Lines 26-27: This sentence requires rephrasing. The FFNNs captured the 'relationship' accurately. Correlation is a method by which we can ascertain this relationship. Also, what do the authors mean by 'as well as precipitation'? Is precipitation well correlated or not? Please clarify.**

: Thank you for pointing this out. The corresponding sentence is corrected as "The FFNNs accurately captured the observed nonlinear RH2m-FRP and precipitation-FRP relationship."

**2. Line 28: Ideally, we expect an inverse relationship between FRP and precipitation. How is this 'excessive' relationship a concern in this context?**

: Sorry for the possible misunderstanding. We deleted the corresponding sentence in the revised manuscript.

**3. Line 34: What kind of fires? Wildfires or agricultural fires? Not all fires cause ecological and socio-economic impacts. Please specify.**

: The corresponding sentence is modified with references as follows "Wildfires are inflicting substantial terrestrial and economical impacts in numerous regions globally (NOAA, 2005; Bowmman et al., 2009).".

**4. Lines 36-37: I see no relevance of this statement here, as the authors have not discussed the ignition factor anywhere in the manuscript.**

: The corresponding sentence is deleted in the revised manuscript.

**5. Line 49: Moisture codes provide no exclusive information about any deceased organic matter.**

: We tried to convey the fact that the initial spread index (ISI) and buildup index (BUI), which combines to obtain the FWI index requires the moisture code as an input. To avoid the confusion, we rephrase the corresponding sentence as follows "…, the moisture codes are provided as an input of the fire behavior indices, such as initial spread index and buildup index to finally calculate the FWI, providing an estimation of fire intensity."

**6. Line 63: It is unclear what the authors are trying to convey here. Also, how is it relevant to the rest**

of the paragraph?

: Sorry for the possible misunderstanding. We modified the corresponding sentence as follows "while the relationship between the meteorological variables and the fire activity varies significant from regions to regions.".

7. Line 65: How can we 'calibrate' sensitivity? Consider changing it to 'estimate/ calculate'.

: The corresponding phrase is modified as follows "To understand the varying sensitivities of wildfire activity to the meteorological variables from different regions,"

8. Line 70: I guess there is a typo. It should be 'fuel' moisture code instead of fire.

: Corrected as a reviewer suggested.

9. Line 116: consider changing 'responsible for' with 'representing'

: It is corrected as a reviewer suggested.

10. Line 127: there is a typo in the equation. It should be yi instead of y1.

: It is i, but looks like 1 or l in the manuscript, once we put the upper brackets notation. To avoid the confusion, it is now modified n.

11. Line 132: Is the title 'Experimental design' 'for section 2.3 suitable? It is rather an explanation of the cross-validation strategy only.

: It is corrected as a reviewer suggested.

12. Line 143-145: Rephrase the sentence. Anomalies were compared and assessed for accuracy.

: Thank you for pointing this out. The corresponding sentence is rephrased as follows "The FRP anomalies, which were calculated by subtracting the estimated daily climatology during 2001–2020 period, were compared and assessed for the FRP estimation accuracy.".

13. Line 161: Please provide a map of FRP climatology in the main or supplementary article. What does the citation here convey?

: We deleted the citation, and provide a map of FRP climatology as Supplementary Fig. S2.

---

## Author Comment (AC2)

**Reviewer #2**

Review of "Regionally optimized fire parameterizations using feed-forward neural networks" by Ham et al.

General comments: In this paper, the authors argue that a deep learning technique, namely feed-forward neural networks (FFNN) has a high skill in predicting the fire radiative power (FRP) in comparison to the traditional fire weather index (FWI) and a linear regression model. The FFNN model is trained with the meteorological variables of 2-m relative humidity (RH2m), precipitation, 2-m temperature, and windspeed). The authors propose that the FFNN-based technique can be a better fire parameterization for the weather models. Overall, this is an interesting manuscript. However, I have some concerns that the authors must address before accepting the manuscript. My concerns are listed below.

**Specific comments:**

1. The manuscript seems to be written as a letter, with only four figures in the main manuscript. I don't think that ESD has a restriction on the number of figures in the main manuscript. So, consider moving some of the supplementary information to the main manuscript. The model architecture needs to be shown in the main manuscript.

: Thank you very much for the suggestion. We agree to the reviewer's comment that some of figures are worthwhile to be moved to main manuscript. We moved 3 Supplementary Figures (model architecture, time-series analysis, LRP result) to the main texts.

2. The training and validation functions are not shown. It is important to show the training and validation curves to see if the model does overfit/underfit. Overall, the methods need more clarity.

: Thank you for pointing this out. We agree with the reviewer's comment that the decrease in both the training and validation loss should be demonstrated that the FFNNs is properly setup. Figure B-1 shows the training and validation loss with respect to the epoch in three specific locations. Total numer of epochs for the traininng is set to 1,000, and early stopping is applied (Raskutti et al., 2014), once the validation loss is not decreased for 100 epoches. It is clearly demonstrated that both the training and validation loss is gradually decreased with the increased epoch; indicating that the FFNNs are successfully formulated. We added Figure B-1, and the related texts as Supplementary Fig. 1 and Line 159-163 of the revised manuscript.

[Figure]

Figure B-1. Training (black) and validation loss (red) with respect to the epoch at the grid point in (a) the middle East (centered at 31°N, 47°E), (b) South America (centered at 9°N, 63°W), and (c) Australia (centered at 13°S, 131°E).

3.The authors compare FFNN with a linear regression model. Why not compare it against an existing parameterization scheme?

: We fully understand the reviewer's question. Actually, we already well followed reviewer's comment as the FWI is one of widely-used parameterizations worldwide, and, our FFNNs is compared to the FWI-based forecasts. Applying the linear regression to the FWI value is just to match the variability between FWI index and the FRP. In other words, we compared the parameterization quality seeking the nonlinearity between the meteorological variable and the FRP in the widely-used meteorology-based fire intensity estimation algorithm (i.e., FWI algorithm) to that seeking the nonlinearity using the neural network weights and the nonlinear activations.

To avoid the confusion by using the terminology of the 'linear regression', we modified the term 'FWI-based linear regression model' to 'FWI-based model' throughout the revised manuscript, and the brief description about the FWI-based model is also modified in the revised manuscript as follows.

Line 128-134 : "A FRP-estimation model based on the FWI was established as a baseline. The FWI is obtained from the daily averages of T2m, RH2m, WS10m, and PRCP, and .... To match the systematic amplitude differences between the FWI and FRP using the different units, a linear regression coefficient of the FRP with respect to the FWI, which was separately calculated for each grid point, is multiplied to produce the FWI-based model."

4. I find the following paper relevant for this study. Zhang et al. (2021) https://doi.org/10.1016/j.ecolind.2021.107735.

: Thank you for the related reference. It has common research interest with ours, but after reading it carefully, we found there are several different points between theirs and ours. 1) their model is to classify the occurrence of the fire, therefore, their model output is simply either 0 (non-fire) or 1 (fire). 2) they did not compare their results with currently used parameterization scheme. 3) they did not provide physical explanations for the improvement in their model. We noted this point with the reference in Line 79-88 of the revised manuscript as follows.

Line 79-88 : "Recently, artificial neutral networks (ANN) have received extensive attention and continue to expand to various application fields. The traditional ANN model with shallow neural networks such as multilayer perceptrons, and convolutional neural networks has been applied to predict the fire probability over the regional domain (Satir et al., 2016), or parameterize the fire occurrence (Zhang et al., 2021) from the meteorological variables. Despite previous literature demonstrating promising accuracy in estimating or predicting fire characteristics, the development of globally applicable ANN-based parameterization is still in its early stages. This is primarily due to the regional idiosyncrasies in the relationships between meteorological variables and fire activity, posing challenges for establishing global implementation."

**References**

Raskutti, G., Wainwright, M. J., & Yu, B. (2014). Early stopping and non-parametric regression: an optimal data-dependent stopping rule. *The Journal of Machine Learning Research*, *15*(1), 335-366.

---

## Author Comment (AC3)

**Regionally optimized fire parameterizations using feed-forward neural networks**

Yoo-Geun Ham[1*], Seung-Ho Nam[2], Geun-Hyeong Kang[2], and Jin-Soo Kim[3*]

[1] Department of Environmental Planning, Graduate School of Environmental Studies, Seoul National University, Seoul, South Korea

[2] Department of Oceanography, Chonnam National University, Gwangju, 61186, South Korea

[3] Low-Carbon and Climate Impact Research Centre, School of Energy and Environment, City University of Hong Kong, Tat Chee Ave, Kowloon Tong, Hong Kong, People's Republic of China

*Correspondence to*: Prof. Yoo-Geun Ham (yoogeun@snu.ac.kr), and Prof. Jin-Soo Kim (jinsoo.kim@cityu.edu.hk)

The fire weather index (FWI) is a widely used metric for fire danger based on meteorological observations. However, due to its empirical formulation based on a specific regional relationship between the meteorological observations and fire intensity, the ability of the FWI to accurately represent global satellite-derived fire intensity observations is limited. In this study, we propose a fire parameterization method using feed-forward neural networks (FFNNs) for individual grids. These FFNNs for each grid point utilize four daily meteorological variables (2-meter relative humidity (RH2m), precipitation, 2-meter temperature, and wind speed) as inputs. The outputs of the FFNNs are satellite-derived fire radiative power (FRP) values. Applying the proposed FFNNs for fire parameterization during the 2001–2020 period revealed a marked enhancement in cross-validated skill compared to parameterization solely based on the FWI. This improvement was particularly notable across East Asia, Russia, the eastern US, southern South America, and central Africa. The sensitivity experiments demonstrated that the RH2m is the most critical variable in estimating the FRP and its regional differences via the FFNNs. Conversely, the FWI-based estimations were primarily influenced by precipitation. The FFNNs accurately captured the observed nonlinear RH2m-FRP and precipitation-FRP relationship compared to that simulated in the FWI-based model.

**Keywords**: fire parameterization, fire radiative power, fire weather index, feed-forward neural networks

**1. Introduction**

Wildfires are inflicting substantial terrestrial and economic impacts in numerous regions globally (Bowman et al., 2009). For example, In 2020, the United States experienced a total of US$16.5 billion in damages due to wildfires, with over 10,000 structures in California alone being damaged or completely destroyed (NOAA, 2021). The 2019-2020 wildfire season in Australia was exceptionally severe, causing smoke-related health costs of AU$1.95 billion, including an estimated 429 premature deaths and over 4,700 hospital visits, a cost nearly nine times the median annual cost of AU$211 million over the previous 19 years (Johnston et al., 2021). Therefore, monitoring and managing the risk of fire incidents at an early stage poses a significant challenge for each country in reducing casualties and economic losses (Vitolo et al., 2019).

As fire propagation is mainly determined by dryness after its ignition, spatially estimating and forecasting dryness enables the monitoring of fire hazards (Bistinas et al., 2014, Abatzoglou and Williams 2016). Facilitating the implementation of emergency measures to curb the expansion of uncontrollable large fires (Di Giuseppe et al., 2016, Bett et al., 2020, Haas et al., 2022). For this reason, in order to prevent fires, various techniques for quantifying and monitoring dryness have been developed and are being used. Indeed, the European Centre for Medium-Range Weather Forecasts (ECMWF) provides the Canadian Forest Fire Weather Index, the Australian McArthur Forest Fire Danger Index, and the Keetch-Byram Drought Index through the European Forest Fire Information System (EFFIS).

Among several operational fire danger indices, the Fire Weather Index (FWI) holds a prominent status as an indicator of potential fire intensity. Developed by the Canadian Forest Fire Danger Rating System (Van Wagner 1974, 1987), the FWI is based on four daily meteorological observations: near-surface air temperature, near-surface air relative humidity, wind speed, and precipitation. Fuel moisture codes are first determined from meteorological data to assign numerical ratings to the moisture content of the forest floor and other deceased organic matter. Afterward, the moisture codes are provided as an input of the fire behavior indices, such as the initial spread index and buildup index, to finally calculate the FWI, providing an estimation of wildfire intensity under given meteorological conditions (Vitolo et al., 2019).

Although this system has been shown to be globally applicable (Bedia et al., 2015, Abatzoglou et al., 2018), it was originally developed for the characterization of evergreen pine stands in forested areas of Canada. Therefore, all links between fire moisture codes and fire behavior indices are optimized and parameterized for eastern

Canada. However, regional fire dynamics vary significantly depending on its unique climatological states (Flannigan et al., 2005, Kim et al., 2019). For example, extensive deforestation fires in the Amazon are attributed to insufficient cumulative precipitation (Le Page et al., 2010), whereas Arctic fire activity is more sensitive to temperature and relevant timing of snowmelt (Kim et al., 2020); however, its regional differences would not be fully considered as the strength of FWI which is originally optimized and derived for physical characteristics of Canadian fire, while the relationship between the meteorological conditions and the fire activity varies significantly from regions to regions.

Artificial neural networks (ANN) have recently received extensive attention and continue expanding to various application fields, including wildfire research. The traditional ANN model with shallow neural networks, such as multilayer perceptron, and convolutional neural networks has been applied to predict the fire probability over the regional domain (Satir et al., 2016), or parameterize the fire occurrence (Zhang et al., 2021) from the meteorological variables. Despite previous literature demonstrating promising accuracy in estimating or predicting fire characteristics, the development of globally applicable ANN-based parameterization is still in its early stages. This is primarily due to the regional idiosyncrasies in the relationships between meteorological variables and fire activity, posing challenges for establishing global implementation.

To understand the varying sensitivities of wildfire activity to the meteorological variables from different regions, our study optimized fire parameterizations with satellite-derived fire radiative power (FRP) datasets based on feed-forward neural networks (FFNNs) in each region with fire activity records. Given that FFNNs follow the same structure and input variables as the FWI, the parameter values linking meteorological observations, fuel moisture code, and fire behavior indices are established for every $1° \times 1°$ resolution grid box via FFNNs, thus foregoing raw parameterizations in the Canadian FWI. In addition to our novel FFNN-based model, we also conducted an in-depth examination of the FWI-based model with FRP for comparative purposes. To quantify the relative contributions of each meteorological parameter to the fire parameterizations, sensitivity experiments were conducted based on climatological values of meteorological observations.

**2. Data and Experimental Design**

2.1. Data

2.1.1. Fire radiative power (FRP)

Given that the FWI was designed to estimate potential fire intensity, our analyses were based on satellite-derived FRP, a metric that represents the rate at which a fire emits energy in the form of thermal radiation. Specifically, daily FRP data was sourced from the Moderate Resolution Imaging Spectroradiometer (MODIS) Collection 6.1 dataset provided by the Fire Information for Resource Management System (FIRMS) (https://firms.modaps.eosdis.nasa.gov/active_fire/) (Giglio et al., 2016). The period of the FRP data spans from 2001 to 2020. The dataset featured a spatial resolution of $1°×1°$ across the entire globe ($0°–360°E$, $90°S–90°N$), with values expressed in megawatts ($10^6$ J s$^{-1}$; MW). It is important to note that although products were generated for both land and ocean areas, we exclusively focused on land values, as FRP is directly associated with fire size and intensity over terrestrial surfaces.

2.1.2. Meteorological observations

Meteorological observations are required as an input of the FWI and the FFNNs for the FRP parameterizations. In this study, we used daily-averaged 2 m air temperature (T2m), 2 m air relative humidity (RH2m), 10 m wind speed (WS10m), and precipitation (PRCP) from ERA5 reanalysis produced by the European Centre for Medium-Range Weather Forecasts (ECMWF) from 2001 to 2020 (Hersbach et al., 2020). The original horizontal resolution was a quarter degree but was interpolated to a $1°×1°$ resolution over the entire globe ($0°–360°E$, $90°S–90°N$).

2.2. Models

2.2.1. FWI-based model

A FRP-estimation model based on the FWI was established as a baseline. The FWI is obtained from the daily averages of T2m, RH2m, WS10m, and PRCP, and the source code to produce the FWI was obtained from the Canadian Forest Service at https://cfs.nrcan.gc.ca/publications/download-pdf/36461. To match the systematic amplitude differences between the FWI and FRP using the different units, a linear regression coefficient of the FRP with respect to the FWI, which was separately calculated for each grid point, is multiplied to produce the FWI-based model. Therefore, the nonlinearity between the meteorological variable and the FRP in the baseline model is purely originated from the procedure to derive the FWI. A cross-validation strategy was adopted for the skill assessment. For more details, please refer to section 2.3.

2.2.2. FFNNs for FRP parameterization

The FFNNs employed for FRP parameterization consist of one input layer, three hidden layers, and one output layer (Figure 1). The input layer comprises four neurons corresponding to daily averages of T2m, RH2m, WS10m, and PRCP at a specific grid point. The output layer, on the other hand, encompasses a single neuron respresening concurrent FRP estimation at the corresponding grid point. Notably, FFNNs are configured individually for each grid point. The first, second, and third hidden layers are composed of 64, 32, and 16 neurons, respectively. Activation functions are implemented utilizing the ReLU function, which is known to be powerful by introducing nonlinearity and solving the vanishing gradient issues (Agarap, 2018).

Techniques such as batch normalization to normalize activations in intermediate layers of deep neural networks (Bjorck et al., 2018), and dropout to prevent an overfitting to the training data by randonmly drop units (Srivastava et al., 2014) with a dropout rate of 0.2, are applied to enhance model robustness.

It should be noted that the meteorological observations serving as input for the FFNNs mirror those employed in the FWI. Thus, any disparities in estimation accuracy between the FFNNs and the FWI-based model solely stem from  the FRP estimation algorithm.

The loss function of the FFNNs is defined as the root-mean-squared difference between the observed FRP (y) and the estimated FRP (ŷ) as follows.

$$\quad \text{Loss} = \sum_{n=1}^{N} (y_n - \widehat{y_n})^2$$

where N denotes the number of training samples. Total numer of epochs for the traininng is set to 1,000, and early stopping is applied (Raskutti et al., 2014), once the validation loss is not decreased for 100 epoches. It is shown that both the training and validation loss is decreased with the increased epoch (Supplementary Fig. S1), indicating that the FFNNs to estimate the FRP are successfully formulated. Similar to the FWI-based model, a cross-validation strategy is adapted for the skill assessment (see section 2.3 for more details).

2.3. Cross-validation strategy for the skill assessment

The performance of both the FFNNs and the FWI-based model was assessed by adopting a cross-validation strategy. The dataset was partitioned into distinct subsets for testing, validation, and training purposes. The testing period was defined by dividing the entire period from 2001 to 2020 into four-year intervals. The validation dataset is defined as the last two years of each four-year interval, whereas the remaining data was used for training. For example, for the 1$^{st}$ Jan. 2001–31$^{st}$ Dec. 2004 test period, the models were trained using a 1$^{st}$ Jan. 2005– 31$^{st}$ Dec. 2018 dataset, whereas the data from 1$^{st}$ Jan. 2019– 31$^{st}$ Dec. 2020 was used for validation. Additional details on the selection of periods for training, validation, and testing are provided in Supplementary

Table S1. After aligning all testing results from multiple sets of experiments with different period for training/validating/testing, the skill in estimating FRP was estimated using both FFNNs and FWI-based models across the 2001–2020 period. We note that evaluating the skill of FFNN against FRP data may lead to an overestimation of its estimation abilities, given that the FFNN is trained using same type of data.

Regrettably, the absence of ground-based observations on fire activity/intensity for the enough period deprives us of the opportunity to cross-reference FFNN-based FRP

estimations with independent observations. The FRP anomalies, which were calculated by subtracting the estimated daily climatology during 2001–2020 period, were compared and assessed for the FRP estimation accuracy.

**3. FRP parameterization using the FFNNs**

Figure 2 illustrates the correlation skill and root-mean-squared error (RMSE) between the observed FRP anomalies from 2001 to 2020 and the FRP anomalies estimated with

FFNNs and the FWI-based model. The correlation skill of the FFNNs exceeded 0.6

over southern China, northern India, southern South America, the eastern US, southern

Africa, western-central Russia, and maritime continents (Figure 2a). In contrast, the correlation skill of the FWI-based model fell below 0.6, with southern China and central

Africa being the only exceptions (Figure 2b). Therefore, the FFNNs consistently exhibited superior correlation skills compared to the FWI-based model over most of the globe (Figure 2c). Notably, the improvement in the correlation skill of the FFNNs was statistically significant at a 95% confidence level, as determined using the method outlined by Zou (2007). This significance was particularly pronounced over East Asia, the entirety of Russia, the eastern US, southern South America, and central Africa.

The RMSE of the FRP estimations tended to be higher over the regions with high

FRP climatology in both models (Supplementary Fig. S2). A clear distinction in the

RMSE emerges upon comparing FFNNs and the FWI-based model; FFNNs demonstrate an RMSE below 1.5 MW across most regions (Figure 2d), while the FWI- based model predominantly registers RMSE values ranging between 1.5 and 1.8 MW

(Figure 2e). Consequently, the global depiction of RMSE differences reveals negative values, illustrating the consistent superiority of FFNNs over the FWI-based model (Figure 2f).

The systematic improvement in the accuracy of the estimated FRP using the

FFNNs was consistently robust when the skill is evaluated after excluding non-wildfire events (i.e., skill evaluation only when observed FRP > 0) (Supplementary Fig. S3) or when considering monthly-averaged FRP anomalies (Supplementary Fig. S4); both estimation of the fire events in daily scale and its interannual variations of the FRPs with FFNNs align more closely with the observed FRPs than the corresponding outputs of the FWI-based model.

To examine the realism of the temporal variation of the estimated FRP in more detail, Figure 3 shows time-series of the yearly-averaged observed and estimated FRP

over Brazil (Figure 3a), Africa (Figure 3b), Siberia (Figure 3c), and Southern China (Figure 3d). The correlation skill across the various regions consistently exhibited higher correlation skill. Interestingly, the daily evolution and its intensity estimation for the record-breaking wildfire events over the Brazil in 2019 (Brando et al., 2020) (Figure

3e), Africa in 2016 (Verhegghen et al., 2016) (Figure 3f), Siberia in 2003 (Huang et al.,

2009) (Figure 3g), and southern China in 2007 (Cao et al., 2017) (Figure 3h) are consistently better estimated in the FFNNs. These findings highlight the superiority of

FFNNs over the FWI-based model not only in estimating overall variations of the fire intensity, and its detailed evolution and intensity of record-breaking wildfire event worldwide by successfully exploring the relationship between the FRP and the meteorological observations.

To identify the main factors that contributed to the superior accuracy of the

FFNNs, sensitivity experiments were conducted by fixing one of the meteorological observations to the daily climatological values (Figure 4); for example, in the RH2m

Clim experiment, the prescribed values of RH2m as an input of the FFNN is the daily climatology during the whole period (i.e., 2001-2020), therefore, its year-to-year variations in the RH2m is removed. Then, the correlation skill difference between the control simulation, that prescribes all input values at the corresponding date, and the
RH2m Clim experiment is calculated to assess the importance of the RH2m in FRP
parameterization. It clearly indicates that the RH2m are the main factors influencing
the accuracy of the FRP estimations in the FFNNs. For example, the correlation skill
difference between the original estimation and the estimation with the climatological
RH2m was close to 0.5 over most of the regions where the original FRP estimations
exhibited high skill (Figure 4a). On the other hand, substituting PRCP with its
climatological value had a negligible impact on the FFNN-based approach (Figure 4b).
Therefore, RH2m was the dominant variable influencing FRP estimations via the
FFNNs method over most of the globe except for a few regions (Figure 4c). The
correlation skill also remained relatively unaffected when daily climatological values
of WS10m, T2m were considered for the FRP estimations using the FFNNs
(Supplementary Fig. S5).

Conversely, when employing the FWI-based model, the alteration in FRP
correlation skill is more pronounced upon substituting PRCP with its daily
climatological values. In regions such as southern China, northern India, southeastern
South America, and the eastern US, the correlation skill decrease is between 0.2 and
0.3 due to this substitution. In contrast, replacing RH2m with its climatology results in
correlation skill differences of less than 0.1 (Figure 4d and 4e). These findings
underscore the importance of PRCP as the meteorological variable with the greatest
influence on FRP estimation using the FWI (Figure 4f). The correlation skill also
remained relatively unaffected when daily climatological values of WS10m, T2m were
considered for the FRP estimations (Supplementary Fig. S6).

To support our arguments that the RH2m is most importance factor in the FFNNs,
we adapted the layer-wise relevance propagation (LRP) technique (Bach et al., 2015;
Barns et al., 2020; Toms et al., 2020), which is widely used for understanding the
relevance of individual features or neurons in neural networks. It provides a so-called
relevance score $R$ for each variable, which linearly decompose the importance of each
input variables as follows by propagating the output value backward toward the input
variables using a chain rule.

$$f(RH2m, PRCP, T2m, WS10m) = R_{RH2m} + R_{PRCP} + R_{T2m} + R_{WS10m}$$
where $f$ is a nonlinear model (i.e., FFNNs) to derive the FRP, and $R_{RH2m}$, $R_{PRCP}$, $R_{T2m}$,
$R_{WS10m}$ is a relevance score of RH2m, PRCP, T2m, and WS10m, respectively. The relative importance of any particular variable to the estimated FRP can be quantified by calculating the degree of the similarity between the output value and the relevance scores. For this purpose, we obtained the relevance score of each variable for each day during the whole testing period (i.e., 2001-2020) and calculated the correlation with the estimated FRP in the FFNNs (Figure 5). This analysis supports our previous notion that the RH2m is the most sensitive factor influencing FRP estimation in FFNNs, with the contributions of other meteorological parameters being comparatively minor.

The dramatic disparity in the relative contributions of RH2m and PRCP between the two models indicates that the factors that drive the predictive performance of the two models were different. Therefore, the relationship between these two key meteorological observations and the FRP estimations will be further explored in the next section to gain insights into the factors that determine the superior performance of the FFNN-based approach.

**4. Physical explanations of the superior performance of FFNNs**

To confirm that the superior performance of the FFNNs is associated with the differences in the relationship between the RH2m and the estimated FRP between the

FFNNs and the FWI-based models, we selected grid points that satisfy the following three conditions: (1) an FRP correlation skill improvement in FFNNs over FWI-based models is greater than a threshold value (i.e., 0.05 in this case), (2) RH2m is the most sensitive variable for FRP estimation in FFNNs (green color in Fig. 2c), and (3) PRCP

is the most sensitive variable in the FWI-based model (blue color in Fig. 2f). A total of

852 grid points were selected based on these criteria, which accounts for approximately

25.1% of total land grid points and 49.7% of total grid points whose correlation skill improvement in the FFNNs is greater than a threshold value of 0.05. The selected grid points are located over southern China, Russia, central Africa, the eastern US, and central-northern South America (Figure 6a). We note that a threshold of 0.1 for correlation skill improvement would not change the general conclusion, which will be discussed in the following paragraph.

Figure 6b-g illustrates the averaged FRP for each RH2m bin with a 10% interval.

[revised manuscript text omitted]

> 3 mm/day, which is almost negligible worldwide (Figure 7f).

As a result, the regression coefficient between the FRP estimation and the PRCP

is systematically greater in the FWI-based model. For observations, the quadratic coefficient is 0.022 MW/(mm/day)$^2$ (black in Figure 7a), and that for the FFNNs 0.023

MW/(mm/day)$^2$ (black in Figure 7c), denoting similar amplitude. On the other hand, the FWI-based model is 0.036 MW/(mm/day)$^2$, which is almost twice to that of the others (black in Figure 7e). This suggests that the FWI-based model is more responsive to changes in PRCP, resulting in a more pronounced FRP decrease with increasing

PRCP. This excessive sensitivity in the estimated FRP to PRCP changes can contribute to the excessive influence of PRCP on the FRP estimations in the FWI-based model, as shown in Figure 4f.

**5. Summary and Discussion**

[revised manuscript text omitted]

Huang, S., Siegert, F., Goldammer, J. G., & Sukhinin, A. I. (2009). Satellite-derived
2003 wildfires in southern Siberia and their potential influence on carbon
sequestration. *International Journal of Remote Sensing*, 30(6), 1479-1492.

Johnston, F.H., Borchers-Arriagada, N., Morgan, G.G., Jalaludin, B., Palmer, A. J.,
Williamson, G. J., and Bowman, D. M. J. S.: Unprecedented health costs of
smoke-related PM2.5 from the 2019–20 Australian megafires, Nat. Sustain., 4,
42–47, https://doi.org/10.1038/s41893-020-00610-5, 2021

Jones, M. W., Abatzoglou, J. T., Veraverbeke, S., Andela, N., Lasslop, G., Forkel, M.,
Smith, A. J. P., Burton, C., Betts, R. A., van der Werf, G. R., Sitch, S., Canadell,
J. G., Santín, C., Kolden, C., Doerr, S. H., and Le Quéré, C.: Global and regional
trends and drivers of fire under climate change, Rev. Geophys., 60,
e2020RG000726, https://doi.org/10.1029/2020RG000726, 2022.

Kim, J. S., Jeong, S. J., Kug, J. S., and Williams, M.: Role of local air-sea interaction
in fire activity over equatorial Asia, Geophys. Res. Lett., 46, 14789–14797,
https://doi.org/10.1029/2019GL085943, 2019.

Kim, J. S., Kug, J. S., Jeong, S. J., Park, H., and Schaepman-Strub, G.: Extensive fires
in southeastern Siberian permafrost linked to preceding Arctic Oscillation, Sci.
Adv., 6, eaax3308, https://doi.org/10.1126/sciadv.aax3308, 2020.

Laurent, P., Mouillot, F., Moreno, M. V., Yue, C., and Ciais, P.: Varying relationships
between fire radiative power and fire size at a global scale, Biogeosciences, 16,
275–288, https://doi.org/10.5194/bg-16-275-2019, 2019.

Le Page, Y., van der Werf, G. R., Morton, D. C., and Pereira, J. M. C.: Modeling fire-
driven deforestation potential in Amazonia under current and projected climate
conditions, J. Geophys. Res.-biogeosciences, 115, G03012,
https://doi.org/10.1029/2009JG001190, 2010.

NOAA (Natl. Ocean. Atmos. Assoc.). 2021. Billion-dollar weather and climate
disasters 2021. Natl. Cent. Environ. Inf., Natl. Ocean. Atmos. Assoc., Washington,
DC. https://www.ncdc.noaa.gov/billions/

Oliveras, I., Anderson, L. O., and Malhi, Y.: Application of remote sensing to
understanding fire regimes and biomass burning emissions of the tropical Andes,
Global Biogeochem. Cy., 28, 480– 496, https://doi.org/10.1002/2013GB004664,
2014.

Papagiannaki, K., Giannaros, T. M., Lykoudis, S., Kotroni, V., & Lagouvardos, K.
(2020). Weather-related thresholds for wildfire danger in a Mediterranean region:
The case of Greece. Agricultural and Forest Meteorology, 291, 108076.

Parks, S. A., Parisien, M.-A., Miller, C., and Dobrowski, S. Z.: Fire Activity and
Severity in the Western US Vary along Proxy Gradients Representing Fuel
Amount and Fuel Moisture, PLoS ONE, 9, e99699,
https://doi.org/10.1371/journal.pone.0099699, 2014.

Rabin, S. S., Melton, J. R., Lasslop, G., Bachelet, D., Forrest, M., Hantson, S., Kaplan,
J. O., Li, F., Mangeon, S., Ward, D. S., Yue, C., Arora, V. K., Hickler, T., Kloster,
S., Knorr, W., Nieradzik, L., Spessa, A., Folberth, G. A., Sheehan, T., Voulgarakis,
A., Kelley, D. I., Prentice, I. C., Sitch, S., Harrison, S., and Arneth, A.: The Fire
Modeling Intercomparison Project (FireMIP), phase 1: experimental and
analytical protocols with detailed model descriptions, Geosci. Model Dev., 10,
1175–1197, https://doi.org/10.5194/gmd-10-1175-2017, 2017.

Raskutti, G., Wainwright, M. J., & Yu, B. (2014). Early stopping and non-parametric
regression: an optimal data-dependent stopping rule. *The Journal of Machine*
*Learning Research*, 15(1), 335-366.

Satir, O., Berberoglu, S., & Donmez, C. (2016). Mapping regional forest fire
probability using artificial neural network model in a Mediterranean forest
ecosystem. *Geomatics, Natural Hazards and Risk*, *7*(5), 1645-1658.
Srivastava, N., Hinton, G., Krizhevsky, A., Sutskever, I., & Salakhutdinov, R. (2014).
Dropout: a simple way to prevent neural networks from overfitting. *The journal*
*of machine learning research*, *15*(1), 1929-1958.
Toms, B. A., Barnes, E. A., and Ebert-Uphoff, I.: Physically interpretable neural
networks for the geosciences: Applications to Earth system variability, J. Adv.
Model. Earth Syst., 12, e2019MS002002, https://doi.org/10.1029/2019ms002002,
2020.
Van Wagner, C. E.: Structure of the Canadian forest fire weather index, Can. For. Serv.
Publ., 1333, 44 pp., 1974.
Van Wagner, C. E.: Development and structure of the Canadian forest fire weather
index system, Canadian Forestry Service, Headquarters, Ottawa, Canada, Forestry
Technical Report, vol. 35, 35 pp., 1987.
Verhegghen, A., Eva, H., Ceccherini, G., Achard, F., Gond, V., Gourlet-Fleury, S., &
Cerutti, P. O. (2016). The potential of Sentinel satellites for burnt area mapping
and monitoring in the Congo Basin forests. *Remote Sensing*, *8*(12), 986.
Vitolo, C., Di Giuseppe, F., Krzeminski, B., and San-Miguel-Ayanz, J.: A 1980–2018
global fire danger re-analysis dataset for the Canadian Fire Weather Indices, Scient.
Data, 6, 190032, https://doi.org/10.1038/sdata.2019.32, 2019.
Ying, L., Cheng, H., Shen, Z., Guan, P., Luo, C., & Peng, X. (2021). Relative humidity
and agricultural activities dominate wildfire ignitions in Yunnan, Southwest China:
Patterns, thresholds, and implications. Agricultural and Forest Meteorology, 307,
108540.
Zhang, G., Wang, M., & Liu, K. (2021). Deep neural networks for global wildfire
susceptibility modelling. *Ecological Indicators*, *127*, 107735.
Zou, G.: Toward using confidence intervals to compare correlations, Psychol. Methods,
12, 399–413, https://doi.org/10.1037/1082-989X.12.4.399, 2007.

[Figure]

Figure 1. Configuration of the FFNNs.

[Figure]

Figure 2. Correlation skill between the observed daily FRP and the estimated FRP values in (a) the FFNNs or (b) FWI-based model during 2001–2020. (c) Difference in the correlation skill in the FFNNs from that in the FWI-base model. RMSEs between the observed daily FRP and the estimated FRP values in (d) the FFNNs, or (e) FWI-based model during 2001–2020. (f) Difference in the RMSE in the FFNNs from that in the FWI-base model. The dots in panels (a) and (b) denote the grid points where the correlation skill exceeds a 95% confidence level based on the t-test; those in panel (c) denote the area whose correlation skill difference is above a 95% confidence level calculated as described by Zou (2007).

[Figure]

Figure 3. Time series of the annually-averaged (left) and daily (right) FRP in the observation (black), FFNNs (red), and FWI-based model (blue) over (a), (b) Brazil (64–40°W, 21–1°S), (c), (d) southern Africa (14–36°E, 18°S–6°N), (e), (f) Siberia (104–134°E, 48–60°N), and (g), (h) southern China (108–120°E, 22°N–30°N). Correlation coefficient between the observation and the FFNNs, and FWI-based model is denoted by the red, and blue in each panel, respectively.

[Figure]

Figure 4. Difference in the correlation skill of the original FRP estimation in the FFNNs from that by prescribing (a) the RH2m or (b) the PRCP as the daily climatological values. (c) Spatial distribution of the meteorological variable where the decrease in correlation is largest by prescribing the climatological value. Panels (d), (e), (f) are the same as (a), (b), and (c) but for the FWI-based model. In panels (c) and (f), 2 m air temperature, PRCP, 10 m wind speed, and RH2m are indicated in red, yellow, green, and purple, respectively.

[Figure]

Figure 5. Correlation skill between the relevance score for each variables derived from layer-wise relevance propagation (LRP) and the estimated FRP in the FFNNs during the 2001–2020 period.

[Figure]

Figure 6. (a) Grid points selected for bin-averaged FRP calculation. Case-averaged FRP with respect to the RH2m with a 10% interval in (b) the observations, (d) FFNNs, and (f) FWI-based model. The figures illustrate the difference in the case-averaged FRP at the upper bin from the lower bin in (c) the observations, (e) FFNNs, and (g) FWI-based model.

[Figure]

Figure 7. Case-averaged FRP with respect to the PRCP with 0.1 mm/day interval in (a) the observations, (c) FFNNs, and (e) FWI-based model. The black line in each panel quadratic shows the fitted line to the quadratic regression, and number in the upper right corner denotes the quadratic coefficients. The figures illustrate the spatial distribution of the case-averaged FRP when the PRCP > 3 mm/day in (b) the observations, (d) FFNNs, and (f) the FWI-based model. The selected areas for the calculation of the bin-averaged values is given in Figure 6a.

---

## Author Comment (AC5)

*Supplements of*

**Regionally optimized fire parameterizations**
**using feed-forward neural networks**

Yoo-Geun Ham[1*], Seung-Ho Nam[2], Geun-Hyeong Kang[2], and Jin-Soo Kim[3*]

[1] Department of Environmental Planning, Graduate School of Environmental Studies, Seoul National University, Seoul, South Korea

[2] Department of Oceanography, Chonnam National University, Gwangju, 61186, South Korea

[3] Low-Carbon and Climate Impact Research Centre, School of Energy and Environment, City University of Hong Kong, Tat Chee Ave, Kowloon Tong, Hong Kong, People's Republic of China

*Correspondence to*: Prof. Yoo-Geun Ham (yoogeun@snu.ac.kr), and Prof. Jin-Soo Kim (jinsoo.kim@cityu.edu.hk)

|  | EXP1 | EXP2 | EXP3 | EXP4 | EXP5 |
|---|---|---|---|---|---|
| Training period | 2001–2014 | 2005–2018 | 2009–2020 & 2001–2002 | 2013–2020 & 2001–2006 | 2017–2020 & 2001–2010 |
| Validation period | 2015–2016 | 2019–2020 | 2003–2004 | 2007–2008 | 2011–2012 |
| Testing period | 2017–2020 | 2001–2004 | 2005–2008 | 2009–2012 | 2013–2016 |

Supplementary Table S1. Periods for the training, validation, and testing of the

FFNNs. Note that every period starts from Jan. 1st and end at Dec. 31st.

[Figure]

Supplementary Fig. S1. Training (black) and validation loss (red) with respect to the epoch at the grid point in (a) the middle East (centered at 31°N, 47°E), (b) South America (centered at 9°N, 63°W), and (c) Australia (centered at 13°S, 131°E).

[Figure]

Supplementary Fig. S2. The spatial distribution of the FRP climatology during 2001-2020 period.

[Figure]

Supplementary Fig. S3. Same as main Fig. 2, but for cases where the observed FRP >
0.

[Figure]

Supplementary Fig. S4. Same as main Fig. 2, but using monthly-averaged FRP.

[Figure]

Supplementary Fig. S5. Difference in the correlation skill of the original FRP

estimation in the FFNNs from that by using (a) the RH2m, (b) PRCP, (c) T2m, and (d)

WS10m as the daily climatological values.

[Figure]

Supplementary Fig. S6. Difference in the correlation skill of the original FRP

estimation in the FWI-based model from that by using (a) the RH2m, (b) PRCP, (c)

T2m, and (d) WS10m as the daily climatological values.